# Structural basis of ECF-σ-factor-dependent transcription initiation

Wei Lin[1,3], Sukhendu Mandal [1], David Degen[1], Min Sung Cho[1], Yu Feng[1], Kalyan Das [2] & Richard H. Ebright [1]

Extracytoplasmic (ECF) σ factors, the largest class of alternative σ factors, are related to primary σ factors, but have simpler structures, comprising only two of six conserved functional modules in primary σ factors: region 2 (σR2) and region 4 (σR4). Here, we report crystal structures of transcription initiation complexes containing *Mycobacterium tuberculosis* RNA polymerase (RNAP), *M. tuberculosis* ECF σ factor σ[L], and promoter DNA. The structures show that σR2 and σR4 of the ECF σ factor occupy the same sites on RNAP as in primary σ factors, show that the connector between σR2 and σR4 of the ECF σ factor–although shorter and unrelated in sequence–follows the same path through RNAP as in primary σ factors, and show that the ECF σ factor uses the same strategy to bind and unwind promoter DNA as primary σ factors. The results define protein-protein and protein-DNA interactions involved in ECF-σ-factor-dependent transcription initiation.

[1] Waksman Institute and Department of Chemistry, Rutgers University, Piscataway, NJ 08854, USA. [2] Rega Institute and Department of Microbiology and Immunology, KU Leuven, 3000 Leuven, Belgium. [3] Present address: School of Medicine and Life Science, Nanjing University of Chinese Medicine, Nanjing 210046, China. Correspondence and requests for materials should be addressed to R.H.E. (email: ebright@waksman.rutgers.edu)

Bacterial transcription initiation is carried out by an RNA polymerase (RNAP) holoenzyme comprising RNAP core enzyme and a σ factor[1]. Bacteria contain a primary σ factor (group-1 σ factor; $\sigma^{70}$ in *Escherichia coli*; $\sigma^A$ in other bacteria) that mediates transcription initiation at most genes required for growth under most conditions and sets of alternative σ factors that mediate transcription initiation at sets of genes required in certain cell types, developmental states, or environmental conditions[1].

Group-1 σ factors contain six conserved functional modules: σ regions 1.1, 1.2, 2, 3, 3/4 linker, and 4 (σR1.1, σR1.2, σR2, σR3, σR3/4 linker, and σR4; Fig. 1a)[1]. σR1.1 plays a regulatory role, inhibiting interactions between free, non-RNAP-bound, σ and DNA. σR1.2, σR2, σR3, and σR4 play roles in promoter recognition. σR2 and σR4 recognize the promoter -10 element and the promoter -35 element, respectively, and σR1.2 and σR3 recognize sequences immediately downstream and immediately upstream, respectively of the promoter -10 element. The σR3/4 linker plays multiple crucial roles[2–11]. The σR3/4 linker connects σR2 to σR4; the σR3/4 linker enters the RNAP active-center cleft, where it interacts with template-strand ssDNA of the unwound transcription bubble, pre-organizing template-strand ssDNA to adopt a helical conformation and to engage the RNAP active center, thereby facilitating initiating-nucleotide binding and de novo transcription initiation; and the σR3/4 linker exits the RNAP active-center cleft by threading through the RNAP RNA exit channel. Before RNA synthesis takes place, the σR3/4 linker serves as a molecular mimic of RNA, or molecular placeholder for RNA, through its interactions with template-strand ssDNA and the RNAP RNA exit channel. As RNA synthesis takes place, the σR3/4 linker then is displaced—off of template-strand ssDNA and out of the RNAP RNA exit channel—driven by steric interactions with the 5′-end of the nascent RNA. The σR3/4 linker must be displaced from template-strand ssDNA during initial transcription; this requirement imposes energy barriers associated with initial-transcription pausing and abortive initiation. The σR3/4 linker must be displaced from the RNAP RNA exit channel during the transition between initial transcription and transcription elongation; this requirement imposes energy barriers that are exploited to trigger promoter escape and to transform the transcription initiation complex into the transcription elongation complex.

Crystal structures of RNAP holoenzyme and transcription initiation complexes containing group-1 σ factors define the protein–protein and protein–nucleic acid interactions involved in group-1-σ-factor-dependent transcription initiation, and extensive biochemical and biophysical characterization defines the protein–protein and protein–nucleic acid interactions and mechanisms involved in group-1-σ-factor-dependent transcription initiation[2,3,6,7,12–19].

Alternative σ factors—with the exception of the alternative σ factor mediating the response to nitrogen starvation ($\sigma^{54}$ in *E. coli*; $\sigma^N$ in other bacteria)[20,21]—are members of the same protein family as group-1 σ factors[1]. Group-2 and group-3 alternative σ factors are closely related in structure to group-1 σ factors, lacking only functional modules σR1.1 (in group-2 σ factors) or σR1.1 and σR1.2 (in group-3 σ factors). The close structural similarity of group-2 and group-3 σ factors to group-1 σ factors, together with crystal structures of transcription initiation complexes containing group-2 σ factors[22], facilitates an understanding of the mechanism of group-2- and group-3-σ-factor-dependent transcription initiation.

Group-4 alternative σ factors—also referred to as "extracytoplasmic σ factors" (ECF σ factors), based on functional roles in response to cell-surface and other extracytoplasmic stresses—are only distantly related to group-1 σ factors and are substantially smaller than group-1 σ factors, lacking four of the six functional modules present in group-1 σ factors (Fig. 1a)[1,23–30]. ECF σ factors comprise only a module related to σR2 (the module that recognizes promoter -10 elements in group-1 σ factors), a module related to σR4 (the module that recognizes promoter -35 elements in group-1 σ factors), and a short σR2/4 linker that has no detectable sequence similarity to the σR3/4 linker of group-1 σ factors. No structural information previously has been reported for RNAP holoenzymes or transcription initiation complexes containing ECF σ factors. In the absence of structural information for ECF σ factors, it has been unclear how ECF σ factors, despite lacking sequences homologous to the σR3/4 linker of group-1 σ factors, are able to connect σR2 and σR4 with an appropriate spacing to recognize promoter -10 and -35 elements, are able to pre-organize the DNA template strand to facilitate initiating-nucleotide binding and de novo transcription initiation; and are able to coordinate entry of RNA into the RNA-exit channel with promoter escape. In addition, in the absence of structural information, and with comparatively limited sequence similarity between σR2 of ECF σ factors and σR2 of group-1 σ factors[1], it has been unclear whether σR2 of ECF σ factors adopts the same fold as σR2 of group-1 σ factors and uses the same strategy to bind and unwind the promoter -10 element as group-1 σ factors.

ECF σ factors are numerically the largest, and functionally the most diverse, alternative σ factors[1,24–30]. Fully 10 of the 13 σ factors in *Mycobacterium tuberculosis* (*Mtb*), the causative agent of tuberculosis, are ECF σ factors: $\sigma^C$, $\sigma^D$, $\sigma^E$, $\sigma^G$, $\sigma^H$, $\sigma^I$, $\sigma^J$, $\sigma^K$, $\sigma^L$, and $\sigma^M$, mediating responses to nutrition depletion, surface stress, temperature stress, oxidative stress, pH stress, growth in stationary phase, and growth in macrophages[31–35]. For example, the *Mtb* ECF σ factor $\sigma^L$ (Supplementary Fig. 1A) mediates the response to oxidative stress and regulates its own synthesis, polyketide-synthase synthesis, cell-wall synthesis, lipid transport, the oxidative state of exported proteins, and virulence[36–38].

In this work, we have determined crystal structures, at 3.3–3.8 Å resolution, of functional transcription initiation complexes comprising *Mtb* RNAP, the *Mtb* RNAP ECF σ factor $\sigma^L$, and nucleic-acid scaffolds corresponding to the transcription bubble and downstream dsDNA of an ECF-σ-factor-dependent RNAP-promoter open complex (*Mtb* RPo-$\sigma^L$) or an RNAP-promoter initial transcribing complex (*Mtb* RPitc-$\sigma^L$) (Table 1; Fig. 1; Supplementary Figs. 1, 2).

## Results

**Structures of *Mtb* RPo-$\sigma^L$ and *Mtb* RPitc-$\sigma^L$.** Structures were determined using recombinant *Mtb* RNAP core enzyme prepared by co-expression of *Mtb* RNAP subunit genes in *E. coli*, recombinant *Mtb* $\sigma^L$, and synthetic nucleic-acid scaffolds based on the sequence of the $\sigma^L$-dependent promoter P-*sigL* (the promoter responsible for expression of the gene encoding $\sigma^L$)[36–38] (Supplementary Figs. 1, 2). Transcription experiments demonstrate that *Mtb* RNAP-$\sigma^A$ holoenzyme (containing the group-1 σ factor $\sigma^A$) does not efficiently perform transcription initiation at the P-*sigL* promoter, whereas *Mtb* RNAP-$\sigma^L$ holoenzyme (containing the ECF σ factor $\sigma^L$) does (Supplementary Fig. 1E). We prepared "downstream-fork-junction" nucleic-acid scaffolds containing P-*sigL* sequences, analogous to the downstream-fork-junction nucleic-acid scaffolds containing consensus group-1-σ-factor-dependent promoter sequences used previously for structural analysis of group-1-σ-factor-dependent transcription initiation (Supplementary Fig. 2, left panels). Because the P-*sigL* transcription start site (TSS) had been mapped only provisionally[36–38], we prepared and analyzed a set of downstream-fork-junction nucleic-acid scaffolds having different lengths—4 nt, 5 nt, 6, or 7 nt—of

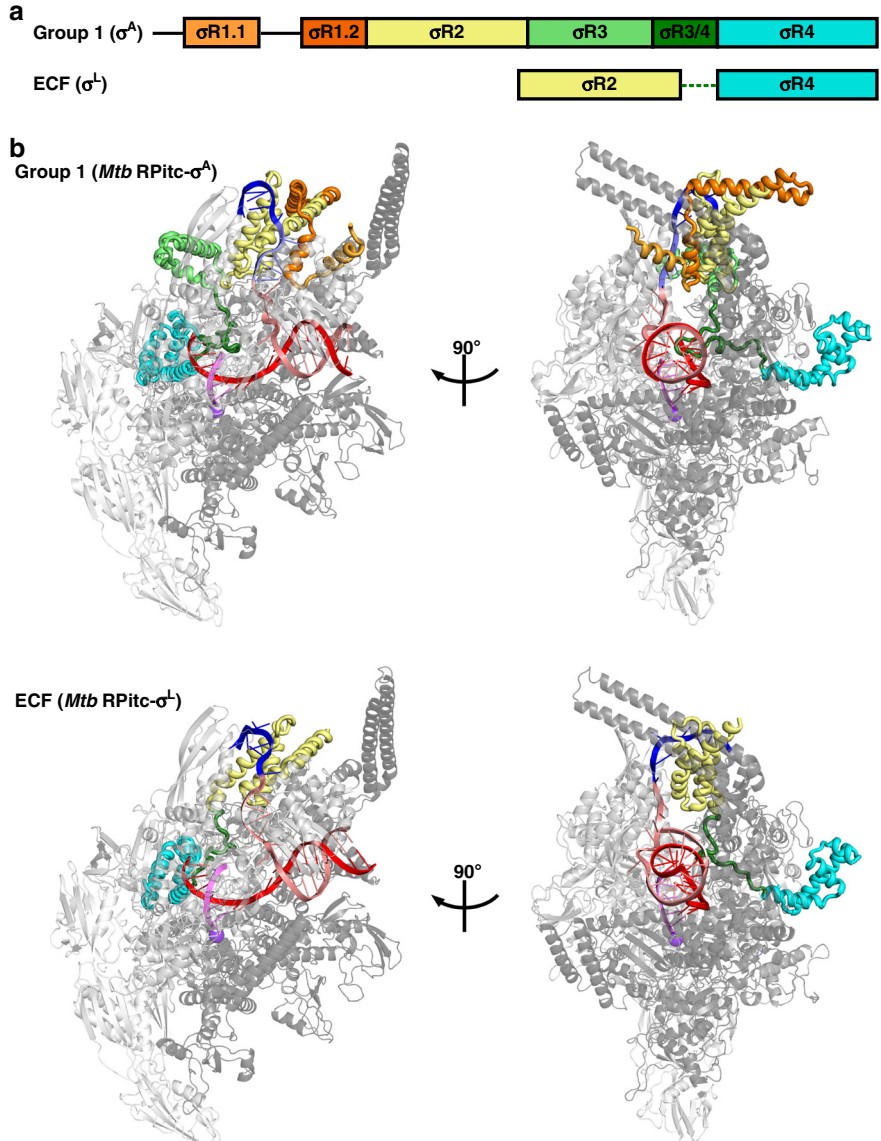

**Fig. 1** Structures of group-1 and ECF σ factors. **a** Structural organization of group-1 (σ$^A$) and ECF (σ$^L$) σ factors. Conserved regions σR1.1, σR1.2, σR2, σR3, σR3/4 linker, and σR4 are in light orange, dark orange, yellow, green, dark green, and cyan, respectively. Dashed line, non-conserved σR2/4 linker present in ECF σ factors. **b** Crystal structures of group-1 (*Mtb* RPitc-σ$^A$; PDB 5UH8) and ECF (*Mtb* RPitc-σ$^L$; PDB 6DVC) transcription initiation complexes (two orthogonal views of each). σ factors are shown in tube representations with conserved regions colored as in (**a**). Gray ribbon, RNAP core enzyme; blue, pink, red, and magenta ribbons, -10 element of DNA nontemplate strand, rest of DNA nontemplate strand, DNA template strand, and RNA product; violet sphere, RNAP active-center Mg$^{2+}$. Other colors are as in (**a**). See Supplementary Figs. 1, 2, and 7

the "spacer" between the P-*sigL* promoter -10 region and down-stream dsDNA (Supplementary Fig. 2, left panels). Transcription experiments indicated that all analyzed nucleic-acid scaffolds were functional in σ$^L$-dependent de novo transcription initiation at the expected TSS (with the initiating nucleotide base-pairing to template-strand ssDNA 2 nt upstream of dsDNA), and σ$^L$-dependent primer-dependent transcription initiation at the expected TSS (with the primer 3′ nucleotide base-pairing to template-strand ssDNA 2 nt upstream of dsDNA), with highest levels of function observed for a spacer length of 6 nt (Supplementary Fig. 1F, G). Robotic crystallization trials identified crystallization conditions yielding high-quality crystals for spacer lengths of 4 nt, 5 nt, or 6 nt (Table 1; Supplementary Fig. 2, center panels). X-ray datasets were collected at synchrotron beam sources, and structures were solved by molecular replacement and refined to 3.3–3.8 Å resolution (Table 1; Supplementary Fig. 2,

right panels). Experimental electron-density maps showed clear density for RNAP, σ$^L$, and nucleic acids (Supplementary Fig. 2, right panels). The resulting structures were essentially identical for nucleic-acid scaffolds having spacer lengths of 4 nt, 5 nt, or 6 nt (Supplementary Fig. 2, right panels). However, map quality was highest for the nucleic-acid scaffold having a spacer length of 6 nt, and therefore subsequent analysis focussed on structures with a spacer length of 6 nt (*Mtb* RPitc5-σ$^L$_sp6). For the nucleic-acid scaffold containing a 6 nt spacer, the translocational state of the transcription complex was experimentally verified by preparation of a scaffold having a single 5-bromo-dU substitution and collection of bromine anomalous diffraction data (Table 1; Supplementary Fig. 2D). The fit of σ$^L$ separately was experimentally verified by preparation of a selenomethionine-labeled σ$^L$ derivative and collection of selenium anomalous diffraction data (Table 1; Supplementary Fig. 2E).

**Table 1 Structure data collection and refinement statistics**

| Structure | Mtb RPitc5-σ$^L$_sp4 | Mtb RPitc5-σ$^L$_sp5 | Mtb RPitc5-σ$^L$_sp6 |
|---|---|---|---|
| PDB code | 6DV9 | 6DVB | 6DVC |
| **Data collection**[a] | | | |
| Source | APS 19-ID | SSRL-9-2 | APS 19-ID |
| Space group | P2$_1$2$_1$2$_1$ | P2$_1$2$_1$2$_1$ | P2$_1$2$_1$2$_1$ |
| Cell dimensions | | | |
| a, b, c (Å) | 143.3, 161.4, 237.7 | 143.7, 160.6, 240.4 | 146.3, 161.5, 240.6 |
| α, β, γ (°) | 90.0, 90.0, 90.0 | 90.0, 90.0, 90.0 | 90.0, 90.0, 90.0 |
| Resolution (Å) | 50.0–3.8 (3.9–3.8) | 50.0-3.8 (3.9–3.8) | 50.0-3.3 (3.4–3.3) |
| Number of unique reflections | 53,020 | 53,728 | 98,083 |
| $R_{merge}$[b] | 0.162 (0.567) | 0.188 (0.706) | 0.175 (0.710) |
| $R_{meas}$ | 0.172 (0.613) | 0.200 (0.752) | 0.184 (0.764) |
| $R_{pim}$ | 0.056 (0.222) | 0.066 (0.246) | 0.055 (0.273) |
| CC$_{1/2}$ (highest resolution shell) | 0.526 | 0.715 | 0.558 |
| $I/\sigma_I$ | 8.6 (2.2) | 7.3 (1.9) | 13 (1.9) |
| Completeness (%) | 96.6 (90.5) | 97.1 (96.5) | 98.9 (99.3) |
| Redundancy | 10.7 (9.9) | 8.2 (8.1) | 10.4 (7.4) |
| Anomalous completeness (%) | N/A | N/A | N/A |
| Anomalous redundancy | N/A | N/A | N/A |
| **Refinement**[a] | | | |
| Resolution (Å) | 50.0–3.8 | 50.0–3.8 | 50.0–3.3 |
| Number of unique reflections | 52,512 | 53,617 | 85,687 |
| Number of test reflections | 2625 | 2691 | 4267 |
| $R_{work}/R_{free}$ | 0.18/0.23 (0.29/0.32) | 0.20/0.24 (0.32/0.35) | 0.19/0.23 (0.34/0.35) |
| Number of atoms | | | |
| Protein | 24,956 | 24,974 | 24,982 |
| Ligand/ion | 3 | 2 | 2 |
| r.m.s. deviations | | | |
| Bond lengths (Å) | 0.002 | 0.002 | 0.002 |
| Bond angles (°) | 0.489 | 0.457 | 0.468 |
| MolProbity statistics | | | |
| Clash score | 7.2 | 6.3 | 6.0 |
| Rotamer outliers (%) | 1.6 | 2.4 | 2.6 |
| Cβ outliers (%) | 0 | 0 | 0 |
| Ramachandran plot | | | |
| Favored (%) | 95.2 | 95.4 | 95.5 |
| Outliers (%) | 0.3 | 0.2 | 0.3 |

| Structure | Mtb [BrU]RPo-σ$^L$_sp6 | Mtb [SeMet15,76]RPo-σ$^L$_sp6 |
|---|---|---|
| PDB code | 6DVD | 6DVE |
| **Data collection**[a] | | |
| Source | APS 19-ID | APS 19-ID |
| Space group | P2$_1$2$_1$2$_1$ | P2$_1$2$_1$2$_1$ |
| Cell dimensions | | |
| a, b, c (Å) | 142.1, 161.5, 239.4 | 142.8, 160.6, 240.2 |
| α, β, γ (°) | 90.0, 90.0, 90.0 | 90.0, 90.0, 90.0 |
| Resolution (Å) | 50.0–3.9 (4.0–3.9) | 50.0–3.8 (3.9–3.8) |
| Number of unique reflections | 44,608 | 50,281 |
| $R_{merge}$[b] | 0.110 (0.768) | 0.178 (>1.000) |
| $R_{meas}$ | 0.121 (0.847) | 0.187 (>1.000) |
| $R_{pim}$ | 0.047 (0.348) | 0.062 (0.608) |
| CC$_{1/2}$ (highest resolution shell) | 0.797 | 0.589 |
| $I/\sigma_I$ | 14.3 (1.6) | 10.2 (1.0) |
| Completeness (%) | 87.4 (68.5) | 92.1 (79.3) |
| Redundancy | 6.0 (4.8) | 9.3 (5.5) |
| Anomalous completeness (%) | 87.4 | 92.1 |
| Anomalous redundancy | 6.0 | 9.3 |
| **Refinement**[a] | | |
| Resolution (Å) | 45.9–3.9 | 46.7–3.8 |
| Number of unique reflections | 37,593 | 40,877 |
| Number of test reflections | 1996 | 1987 |
| $R_{work}/R_{free}$ | 0.22/0.24 (0.25/0.26) | 0.20/0.24 (0.25/0.30) |
| Number of atoms | | |
| Protein | 24,743 | 24,782 |
| Ligand/ion | 2 | 4 |
| r.m.s. deviations | | |
| Bond lengths (Å) | 0.002 | 0.002 |
| Bond angles (°) | 0.491 | 0.492 |
| MolProbity statistics | | |
| Clash score | 6.6 | 6.8 |
| Rotamer outliers (%) | 2.6 | 1.7 |
| Cβ outliers (%) | 0 | 0 |
| Ramachandran plot | | |
| Favored (%) | 95.3 | 95.0 |
| Outliers (%) | 0.3 | 0.3 |

[a]Data for the highest resolution shell are presented in parentheses
[b]$R_{merge}$ values for 6DVB, 6DVC, 6DVD, and 6DVE reflect an anisotropic component

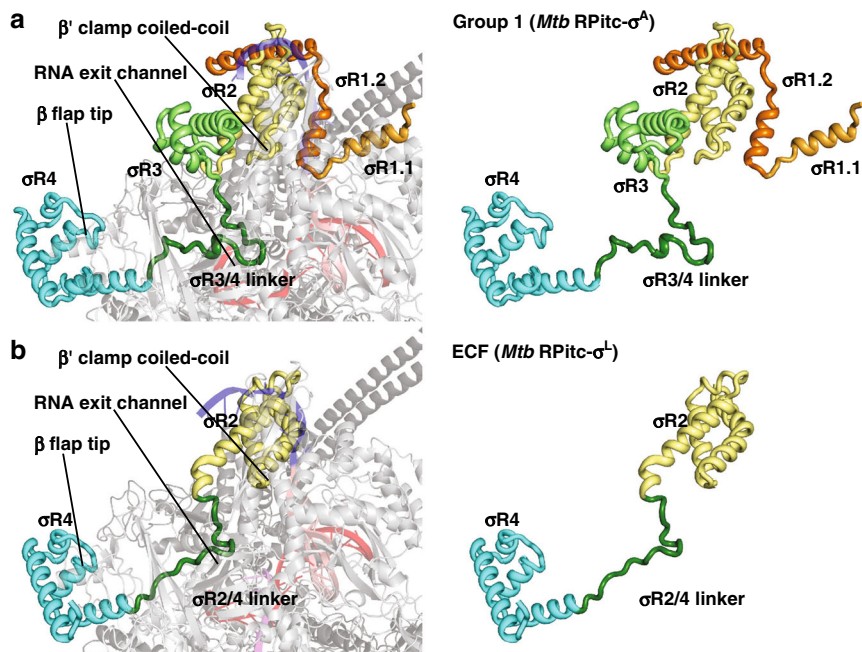

**Fig. 2** Protein–protein interactions between group-1 and ECF σ factors and RNAP core enzyme. **a** Protein–protein interactions by group-1 (σ^A) σ factor. **b** Protein–protein interactions by ECF (σ^L) σ factor. Colors are as in Fig. 1. See Supplementary Fig. 7

**Interactions between ECF σ factor and RNAP.** The structural organization of the ECF σ^L-factor-dependent transcription initiation complex is unexpectedly similar to that of a group-1 σ^A-factor-dependent transcription initiation complex (Figs. 1b and 2). σ2 and σ4 of σ^L occupy the same positions on RNAP, and make the same interactions with RNAP, as σ2 and σ4 of σ^A factor (Fig. 2). Despite the smaller size of the connector between σ2 and σ4 in σ^L as compared to σ^A (20 residues vs. 84 residues if one includes σ3; 20 residues vs. 28 residues if one does not include σ3; Supplementary Fig. 1A), the connector in σ^L spans the full distance between the σ2 and σ4 binding positions on RNAP and follows a path through RNAP similar to that of the connector in σ^A (Fig. 2). Thus, the σ^L σ2/4 linker, like the σ^A σ3/4 linker[2–4,6,7,12–19], first enters the RNAP active-center cleft and approaches the RNAP active center, and then makes a sharp turn and exits the RNAP active-center cleft through the RNAP RNA-exit channel.

Inside the RNAP active-center cleft, the σ^L σ2/4 linker, like the σ^A σ3/4 linker[6,7,14–19], makes direct interactions with template-strand ssDNA nucleotides of the unwound transcription bubble (Figs. 2–4, Supplementary Fig. 3A). The interactions of the σ^L σ2/4 linker with template-strand ssDNA include a direct H-bonded interaction of σ^L Ser96 with a Watson–Crick H-bonding atom of the template-strand nucleotide at promoter position -5 (Fig. 3b; Supplementary Fig. 3A, bottom). The interactions of the σ^L σ2/4 linker with template-strand ssDNA are similar to, but less extensive than, those of the σ^A σ3/4 linker with template-strand ssDNA, which include direct H-bonded interactions of σ^A Asp432 and Ser433 with Watson–Crick H-bonding atoms of template-strand ssDNA nucleotides at promoter positions -4 and -3 (Fig. 3a; Supplementary Fig. 3A).

In the case of the group-1 σ factor, σ^A, the interactions between this segment of the σR3/4 linker and template-strand ssDNA pre-organize template-strand ssDNA to adopt a helical conformation and to engage the RNAP active-center nucleotide-addition site[6], thereby facilitating initiating-nucleotide binding and de novo initiation[2,5,6,8]. The similarity of the interactions made by the ECF σ factor, σ^L, suggests that ECF σ factors likewise pre-organize

template-strand ssDNA and facilitate initiating-nucleotide binding and de novo initiation.

In the case of the group-1 σ factor, σ^A, the interactions between this segment of the σR3/4 linker and template-strand ssDNA must be broken, and this segment of the σR3/4 linker must be displaced, when the nascent RNA reaches a length >4 nt during initial transcription, and this requirement for breakage of interactions and displacement is thought to impose an energy barrier that results in, or enhances, abortive initiation[2,5,67,8] and initial-transcription pausing[9–11]. The similarity of the interactions made by the ECF σ factor, σ^L, suggests that ECF σ factors likewise have a similar requirement for displacement of a linker segment during initial transcription—in this case, when the nascent RNA reaches a length of >5 nt (Supplementary Fig. 3B)—and that this similar requirement imposes an energy barrier that results in, or enhances, abortive initiation and initial-transcription pausing. Consistent with this hypothesis, transcription and transcript-release experiments indicate that Mtb RNAP-σ^L holoenzyme efficiently performs abortive initiation, producing and releasing short abortive RNA products (Supplementary Fig. 3C).

The five C-terminal residues of the σ^L σ2/4 linker, like the ten C-terminal residues of the σ^A σ3/4 linker, exit the RNAP active-center cleft and connect to σ4 by threading through the RNAP RNA-exit channel (Fig. 2). In the case of the group-1 σ factor, σ^A, the C-terminal segment of the σR3/4 linker must be displaced from the RNA-exit channel when the nascent RNA reaches a length of 11 nt at the end of initial transcription and moves into the RNA-exit channel, and this displacement is thought to alter interactions between σR4 and RNAP, thereby triggering promoter escape and transforming the transcription initiation complex into a transcription elongation complex[2–4]. The similarity of the threading through the RNAP RNA-exit channel by the ECF σ factor, σ^L, suggests that ECF σ factors have a similar requirement for displacement of a linker C-terminal segment and have a similar mechanism of promoter escape and transformation from transcription initiation complexes into transcription elongation complexes.

In its interactions with template-strand ssDNA and the RNAP RNA-exit channel, the σ^L σ2/4 linker, like the σ^A

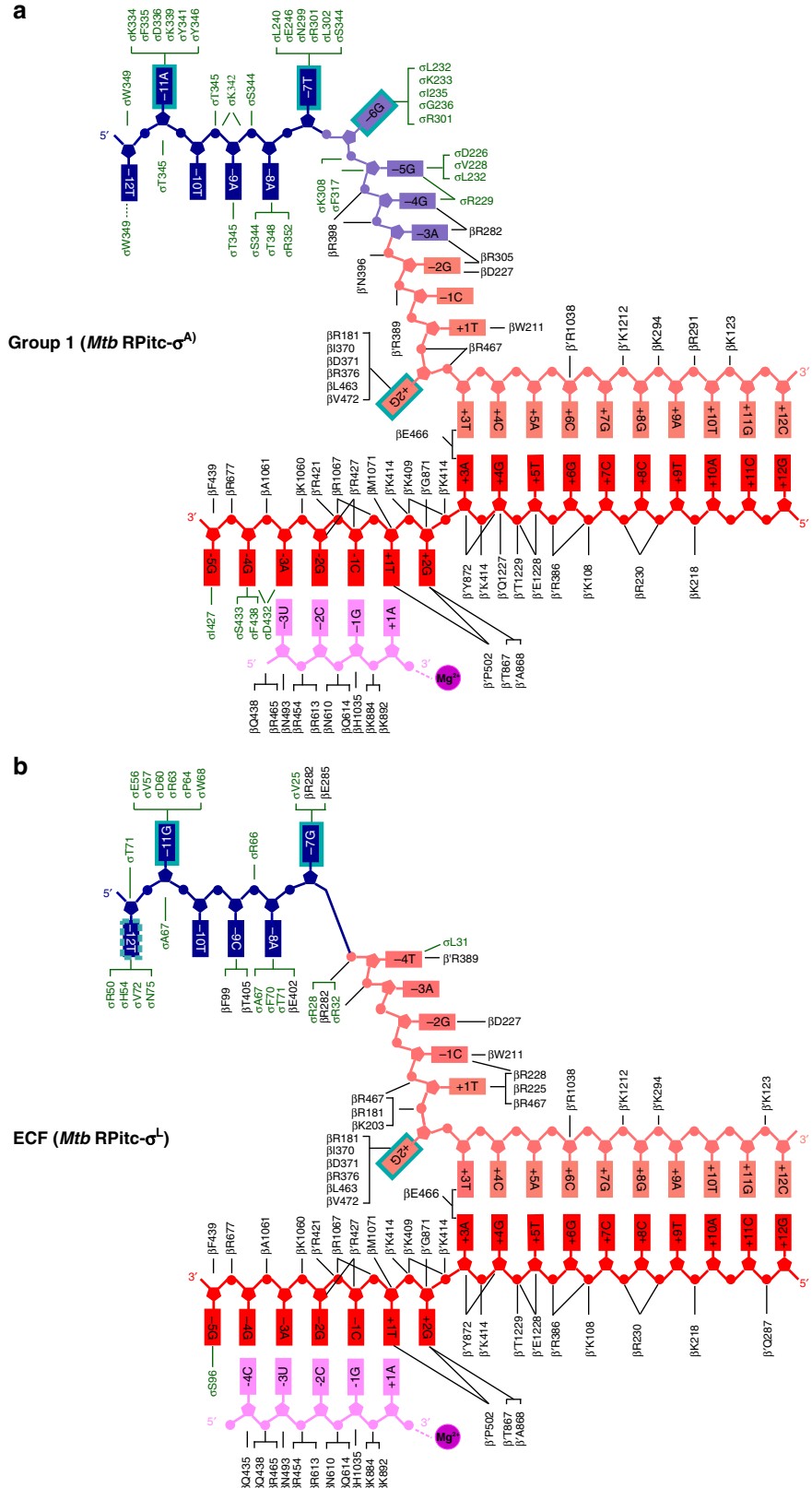

**Fig. 3** Protein–nucleic acid interactions by group-1 and ECF σ factors: summary. **a** Summary of protein–nucleic acid interactions in *Mtb* RPitc-σ^A. Black residue numbers and lines, interactions by *Mtb* RNAP; green residue numbers and lines, interactions by *Mtb* σ^A; blue, -10 element of DNA nontemplate strand; light blue, discriminator element of DNA nontemplate strand; pink, rest of DNA nontemplate strand; red, DNA template strand; magenta, RNA product; violet circle, RNAP active-center Mg²⁺; cyan boxes, bases unstacked and inserted into protein pockets. Residues are numbered as in *Mtb* RNAP. **b** Summary of protein–nucleic acid interactions in *Mtb* RPitc-σ^L. Green residue numbers and lines, interactions by *Mtb* σ^L. Other colors are as in (**a**). See Supplementary Figs. 3–5 and 7

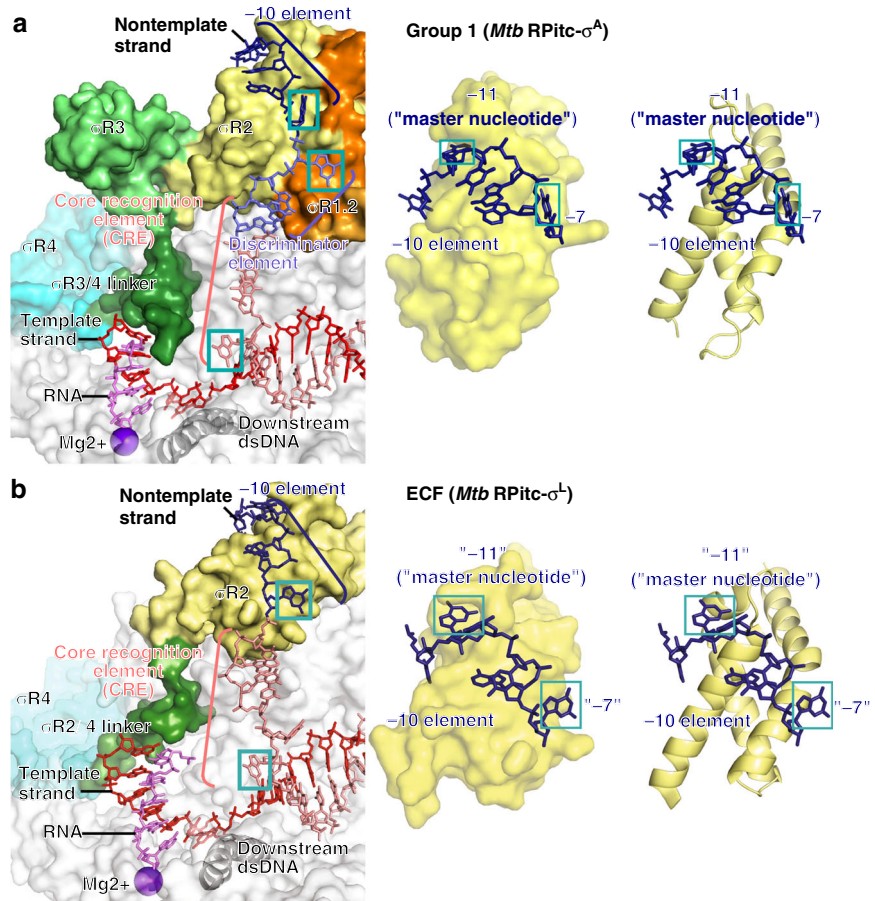

**Fig. 4** Protein–nucleic acid interactions by group-1 and ECF σ factors: interactions with transcription bubble. **a** Left: interactions of *Mtb* RNAP and σ[A] with transcription-bubble nontemplate strand, transcription-bubble template strand, and downstream dsDNA. Right: interactions of σ[A] σR2 with σ[A]-dependent promoter -10 element. For promoter positions -11 ("master nucleotide")[40] and -7, bases are unstacked and inserted into pockets (cyan boxes). Colors are as in Figs. 1–3. **b** Left: interactions of *Mtb* RNAP and σ[L] with transcription-bubble nontemplate strand, transcription-bubble template strand, and downstream dsDNA. Right: interactions of σ[L] σR2 with σ[L]-dependent promoter -10 element. For two promoter positions, here designated "-11" ("master nucleotide") and "-7", by analogy to corresponding nucleotides in group-1-σ-factor complex (panel **a**), bases are unstacked and inserted into pockets (cyan boxes). For one additional nucleotide, here designated "-12", the base also appears to be unstacked and inserted into a pocket (dashed cyan box). See Supplementary Figs. 4, 5 and 7

σR3/4 linker[2–4], appears to serve as a molecular mimic, or a molecular placeholder, for nascent RNA, making interactions with template-strand ssDNA and the RNAP RNA-exit channel in early stages of transcription initiation that subsequently, in late stages of transcription initiation and in transcription elongation, are made by nascent RNA. The σ[L] σR2/4 linker and the σ[A] σR3/4 linker, both have net negative charge (Supplementary Fig. 1A), and both employ extended conformations (fully extended for the σ[L] σR2/4 linker; largely extended for the σ[A] σR3/4 linker; Fig. 2) to interact with template-strand ssDNA and the RNAP RNA-exit channel, consistent with function as molecular mimics of a negatively charged, extended nascent RNA. Nevertheless, the σ[L] σR2/4 linker and the σ[A] σR3/4 linker exhibit no detectable sequence similarity (Supplementary Fig. 1A) and no detailed structural similarity (Fig. 2). We conclude that the σR2/4 linker of an ECF σ factor and the σR3/4 linker of a group-1 σ factor provide an example of functional analogy in the absence of structural homology.

**Interactions between ECF σ factor and promoter -10 element.** The structure reveals the interactions between the ECF σ factor, σ[L], and promoter DNA that mediate recognition of the promoter

-10 element (Figs. 3–5; Supplementary Fig. 4). The σ[L] conserved module σR2, like the σ[A] conserved module σR2, mediates recognition of the promoter -10 element through interactions with nontemplate-strand ssDNA in the unwound transcription bubble (Figs. 3 and 4). In the case of the group-1 σ factor, σ[A], a crucial aspect of recognition of the promoter -10 element is unstacking of nucleotides, flipping of nucleotides, and insertion of nucleotides into protein pockets at two positions of the σ[A]-dependent promoter[6,39]: i.e., position -11 (referred to as the "master nucleotide", based on its especially important role in promoter recognition)[40] and position -7 (Figs. 3a and 4a; Supplementary Fig. 4). The ECF σ factor, σ[L], unstacks nucleotides, flips nucleotides, and inserts nucleotides into protein pockets at the corresponding positions of the σ[L]-dependent promoter (here designated positions "-11" and "-7"; Figs. 3b, 4b, and 5; Supplementary Fig. 4) and also unstacks and inserts a nucleotide into a protein pocket at one additional position of the σ[L]-dependent promoter (position "-12"; Figs. 4b and 5).

RNAP σ[L] holoenzyme unstacks, flips, and inserts into a protein pocket a guanosine at position "-11" of the σ[L]-dependent promoter, making extensive interactions with the base moiety of the guanosine, including multiple direct H-bonded interactions with Watson–Crick H-bonding atoms (Figs. 3, 4b, and 5;

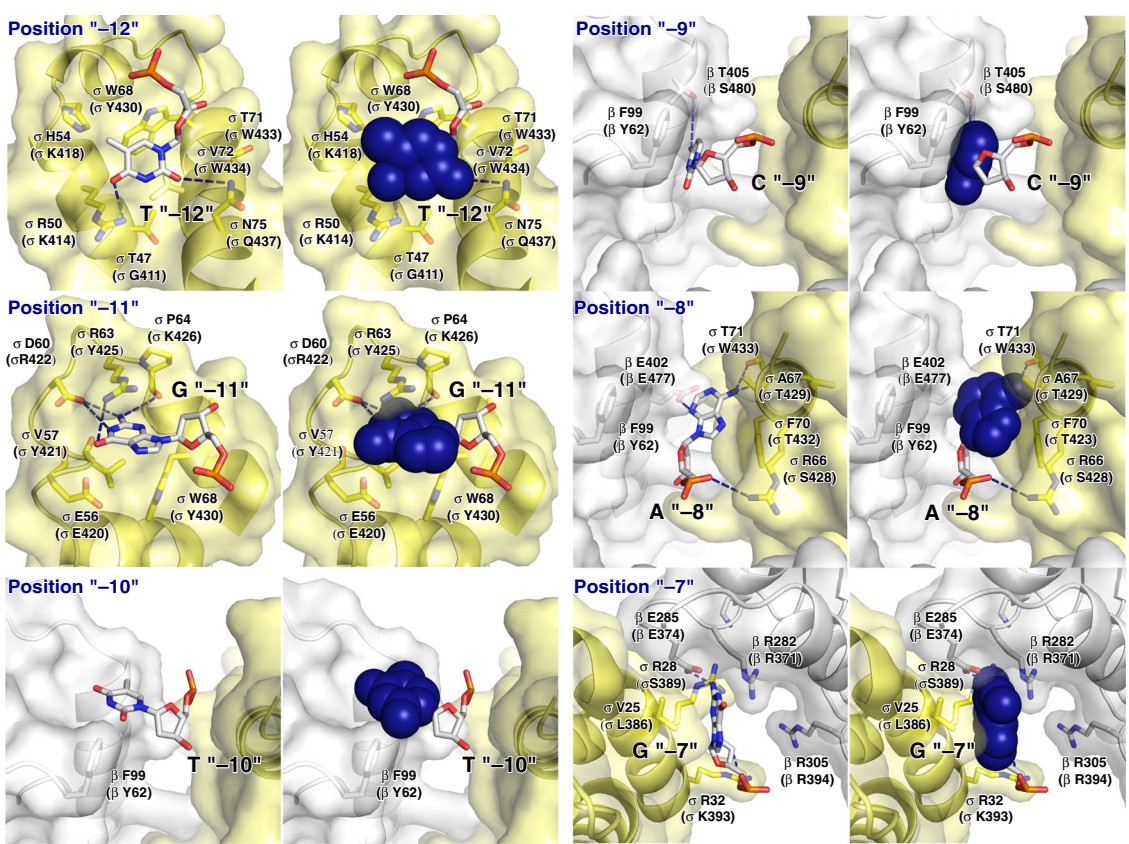

**Fig. 5** Recognition by *Mtb* σ$^L$ of σ$^L$-promoter -10 element: interactions with nontemplate strand positions "-12" through "-7". For each promoter position, left subpanel shows DNA nucleotides in stick representation to highlight individual protein-nucleotide interactions, and right subpanel shows DNA nucleotide base moieties in space-filling representation to highlight protein-base steric complementarity. Yellow surfaces, solvent-accessible surfaces of *Mtb* σ$^L$ σR2; gray surfaces, solvent-accessible surfaces of *Mtb* RNAP β subunit; dark blue surfaces, van der Waals surfaces of bases of σ$^L$-promoter -10 element; yellow ribbons, *Mtb* σ$^L$ σR2 backbone; gray ribbons, *Mtb* RNAP β subunit backbone; yellow, yellow-blue, and yellow-red stick representations, σ$^L$ carbon, nitrogen, and oxygen atoms, respectively; white, blue, red, and orange stick representations, DNA carbon, nitrogen, oxygen, and phosphorous atoms, respectively; blue dashed lines, H-bonds. Residues are numbered as in *Mtb* RNAP and σ$^L$, and, in parentheses, as in *E. coli* RNAP and σ$^{70}$. See Supplementary Fig. 4

Supplementary Fig. 4A). The interactions between σ$^L$ and guanosine at position "-11" of the σ$^L$-dependent promoter are similar to the interactions between RNAP σ$^A$ holoenzyme and adenosine at position -11 of the σ$^A$-dependent promoter -10 element, including, in particular, similar stacking interactions of σ$^L$ aromatic amino acid Trp68 with guanosine and of corresponding σ$^A$ aromatic amino acid Tyr436 with adenosine (Supplementary Fig. 4A). The different specificities—guanosine at position "-11" for σ$^L$ vs. adenosine at position -11 for σ$^A$— arise from differences in the H-bond-donor/H-bond-acceptor character of atoms forming the floors of the relevant protein pockets of σ$^L$ and σ$^A$, with H-bonding complementarity to guanosine in σ$^L$ and H-bonding complementarity to adenosine in σ$^A$ (Supplementary Fig. 4A).

RNAP σ$^L$ holoenzyme also unstacks, flips, and inserts into a protein pocket a guanosine at position "-7" of the σ$^L$-dependent promoter, making extensive interactions with the base moiety of the guanosine, including a direct H-bonded interaction with a Watson–Crick H-bonding atom (Figs. 3, 4b, and 5; Supplementary Fig. 4B). These interactions are similar in location to, but different in detail from, the interactions made by RNAP σ$^A$ holoenzyme with thymidine at position -7 of the σ$^A$-dependent promoter (Fig. 4; Supplementary Fig. 4B). The differences in detail arise from the fact that σ$^L$ does not contain conserved module σR1.2. In the case of σ$^L$, the interactions involve residues of σR2 and residues of RNAP β subunit, with the base moiety of the

guanosine at position "-7" being inserted into a cleft between σR2 and β (Fig. 5; Supplementary Fig. 4B). In contrast, in the case of σ$^A$, the interactions involve residues of σR2 and residues of σR1.2, with the base moiety of the thymidine at position -7 being inserted into a cleft between σR2 and σR1.2 (Supplementary Fig. 4B).

RNAP σ$^L$ holoenzyme also appears to unstack and insert into a protein pocket a thymidine at position "-12" of the σ$^L$-dependent promoter (Figs. 4b and 5), placing one face of the base moiety of the thymidine in a shallow surface pocket, in position to make a direct H-bonded interaction with a Watson–Crick atom (Fig. 5). The interaction with an unstacked nucleotide inserted into a protein pocket implies that position "-12" of the σ$^L$-dependent promoter must be ssDNA in RPo-σ$^L$ and RPitc-σ$^L$, and thus that the transcription bubble must extend to position "-12" in RPo-σ$^L$ and RPitc-σ$^L$. This interaction does not have a counterpart in the σ$^A$-dependent transcription initiation complex, in which position -12 of the σ$^A$-dependent promoter is dsDNA and in which the transcription bubble extends only to position -11[15–19].

In addition to these potential specificity-determining interactions with unstacked nucleotides inserted into protein pockets, RNAP σ$^L$ holoenzyme makes potentially specificity-determining interactions with positions "-9" and "-8" of the σ$^L$-dependent promoter (Fig. 5). RNAP σ$^L$ holoenzyme makes a direct H-bonded interaction, through RNAP β subunit, with a Watson–Crick atom of the base moiety of cytidine at position

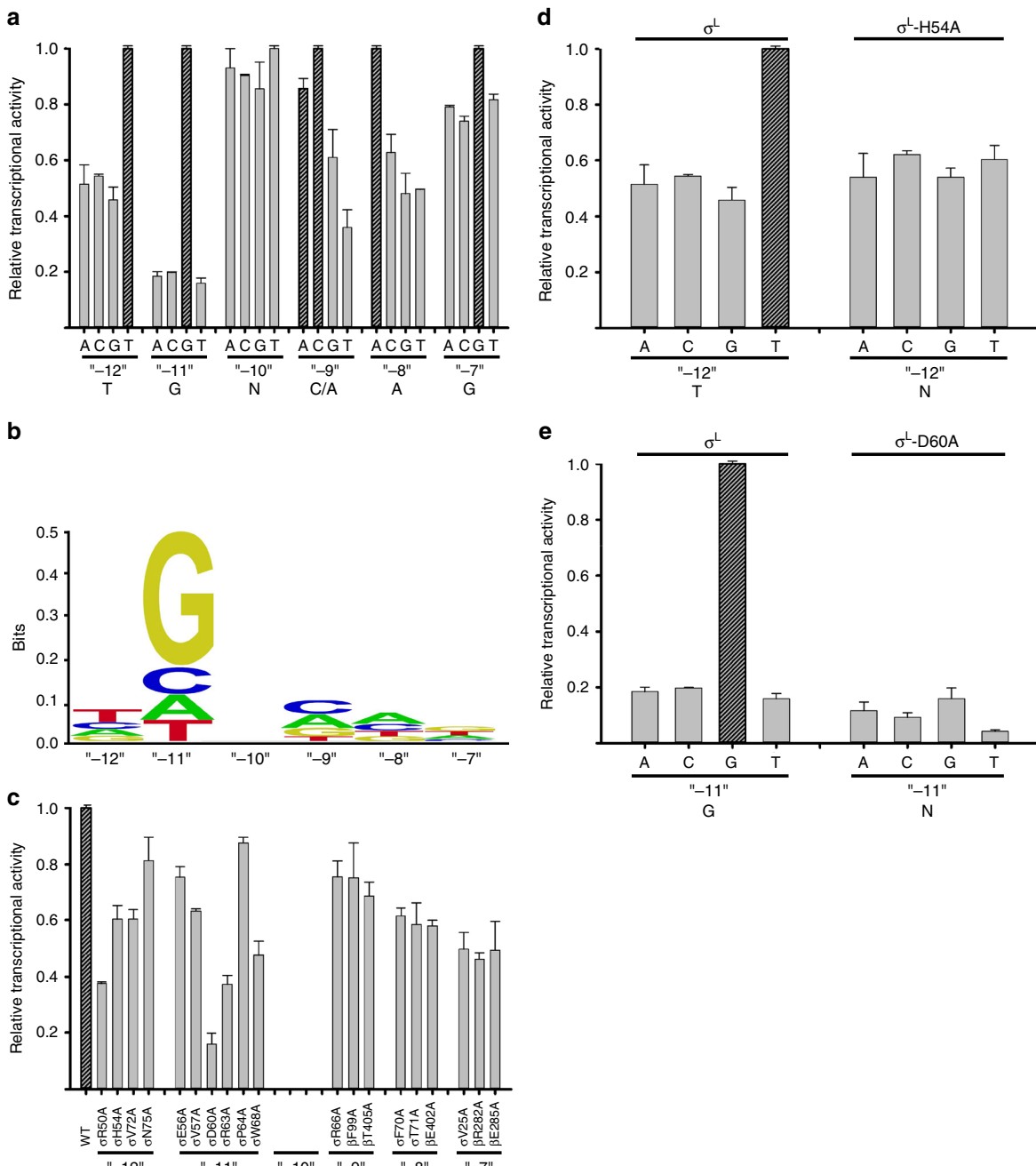

**Fig. 6** Recognition by *Mtb* σ[L] of σ[L]-promoter -10 element: experimental data. **a** Systematic-substitution experiments defining σ[L]-dependent promoter -10-element consensus sequence. Relative transcriptional activities of derivatives of σ[L]-dependent promoter P-*sigL* having all possible single base-pair substitutions at each position of promoter -10 element, "-12" through "-7". Inferred consensus nucleotides are shown at the bottom, and data for inferred consensus nucleotides are hatched. Error bars, SE ($N = 3$). **b** Sequence logo for σ[L]-promoter -10-element consensus sequence [generated using transcription data from (**a**) and enoLOGOS[71] (http://biodev.hgen.pitt.edu/enologos/); input setting "energy (2)" and weight-type setting "probabilities"]. **c** Alanine-scanning experiments[41] demonstrating functional importance of observed amino acid-base interactions in recognition of σ[L]-promoter -10 element. Effects on transcription of alanine substitutions of σ[L] amino acids that contact σ[L]-dependent promoter -10 element, positions "-12" through "-7" (identities of contacting amino acids from Figs. 3 and 5). **d, e** Loss-of-contact experiments[42-45] indicating that σ[L] residues His54 and Asp60 determine specificity at position "-12" and "-11", respectively. Left: transcriptional activity with wild-type σ[L] for all possible single base-pair-substitutions at indicated position (strong specificity for consensus base pair). Right: transcriptional activity of σ[L] derivatives having alanine substitutions (no specificity for consensus base pair). Error bars, SE ($N = 3$). See Supplementary Fig. 6. Source data are provided as a Source Data file

"-9" (Fig. 5) and makes two direct H-bonded interactions, through σR2 and RNAP β subunit, with a Watson–Crick atom of the base moiety of adenosine at position "-8" (Fig. 5).

Biochemical experiments assessing effects of all possible single-base-pair substitutions at each position of the P-*sigL* promoter -10 region confirm the functional significance of the positions contacted in the crystal structure (positions "-12", "-11", "-9", "-8", and "-7"; Fig. 6a), confirm the sequence preferences at these positions inferred from the crystal structure (Fig. 6a), and yield a revised consensus sequence for the σ[L]-dependent -10 element of

T$_{-12}$-G$_{-11}$-N$_{-10}$-C/A$_{-9}$-A$_{-8}$-G$_{-7}$ (Fig. 6b). The revised consensus sequence for the $\sigma^L$-dependent -10 element matches the literature consensus sequence[33–36] in its first four positions (T$_{-12}$-G$_{-11}$-N$_{-10}$-C/A$_{-9}$) and extends the literature consensus sequence for two additional positions (A$_{-8}$-G$_{-7}$). Consistent with the observed extensive network of H-bonded interactions involving position "-11G" (Figs. 3 and 5; Supplementary Fig. 4A), specificity is observed to be strongest at position "-11" (Fig. 6a, b). Three of four characterized *Mtb* $\sigma^L$-dependent promoters[36,37] match the consensus sequence of Fig. 6b at position "-11" (P-*sigL*, P-*pks10*, and P-Rv1139c); two of four match at newly defined position "-8" (P-*sigL* and P-*pks10*), and two of four match at newly defined position "-7" (P-*sigL* and P-Rv1139c).

"Alanine-scanning" experiments[41], in which residues of $\sigma^L$ that contact -10-element nucleotides in the crystal structure are substituted with alanine and effects on $\sigma^L$-dependent transcription are quantified, confirm the functional significance of the observed interactions (Fig. 6c).

"Loss-of-contact" experiments[42–45], in which residues of $\sigma^L$ that contact -10-element nucleotides in the crystal structure are substituted with alanine and effects on specificity at the contacted positions are quantified, confirm that $\sigma^L$ His54 contributes to specificity for thymidine at position "-12" (Fig. 6d) and that $\sigma^L$ Asp60 contributes to specificity for guanosine at position "-11" (Fig. 6e). In the crystal structure, $\sigma^L$ His54 makes a van der Waals interaction with the 5′-methyl group of the base moiety of thymidine at position "-12" (Fig. 5); in loss-of-contact experiments, substitution of His54 by alanine eliminates specificity for thymidine at position "-12" (Fig. 6d). In the crystal structure, $\sigma^L$ Asp60 makes an H-bonded interaction with Watson–Crick atoms of the base moiety of guanosine at position "-11" (Fig. 5; Supplementary Fig. 4A); in loss-of-contact experiments, substitution of Asp60 by alanine eliminates specificity for guanosine at position "-11" (Fig. 6e).

**Interactions between ECF σ factor and promoter CRE.** The structure reveals the interactions between RNAP $\sigma^L$ holoenzyme and nontemplate-strand ssDNA downstream of the promoter -10 element in the ECF $\sigma^L$-dependent transcription initiation complex (Fig. 3b; Supplementary Fig. S5). In the case of group-1-σ-factor-dependent transcription initiation complexes, sequence-specific interactions occur between RNAP β subunit and a 6 nt segment of nontemplate-strand ssDNA downstream of the promoter -10 element referred to as the "core recognition element" (CRE; positions −6 through +2)[6,19,46]. These interactions include, most notably, (1) stacking of a tryptophan residue of RNAP β subunit on the base moiety of thymidine at nontemplate-strand position +1 (Supplementary Fig. 5A), and (2) unstacking, flipping, and insertion into a protein pocket, formed by the RNAP β subunit, of the guanosine at nontemplate-strand position +2 (Figs. 3 and 4a; Supplementary Fig. 5B). The identical interactions occur in the ECF $\sigma^L$-dependent transcription initiation complex (Figs. 3 and 4b; Supplementary Fig. 5).

Biochemical experiments assessing effects of all possible base-pair substitutions at positions downstream of the P-*sigL* promoter -10 element (positions −4 through +2) confirm the functional significance of the interactions in the crystal structure with thymidine at position +1 and guanosine at position +2 (Supplementary Fig. 6A) and yield a CRE consensus sequence for an ECF $\sigma^L$-dependent transcription initiation complex (Supplementary Fig. 6B) similar to the CRE consensus sequence for a group-1-σ-factor-dependent transcription initiation complex[6,46]. Three of four characterized *Mtb* $\sigma^L$-dependent promoters[36,37] match the CRE consensus sequence of Supplementary Fig. 6B at position +2, the position at which a guanosine

is unstacked, flipped, and inserted into a protein pocket (P-*sigL*, P-*pks10*, and P-Rv1139c).

**Discussion**

Our structural results show that: (1) σR2 and σR4 of an ECF σ factor $\sigma^L$ adopt the same folds and interact with the same sites on RNAP as σR2 and σR4 of a group-1 σ factor (Figs. 1 and 2); (2) the connector between σR2 and σR4 of ECF σ factor $\sigma^L$ enters the RNAP active-center cleft to interact with template-strand ssDNA and then exits the RNAP active-center cleft by threading through the RNAP RNA-exit channel in a manner functionally analogous—but not structurally homologous—to the connector between σR2 and σR4 of a group-1 σ factor (Figs. 1 and 2; Supplementary Fig. 3); (3) ECF σ factor $\sigma^L$ recognizes the -10 element of a $\sigma^L$-dependent promoter by unstacking nucleotides and inserting nucleotides into protein pockets at three positions of the transcription-bubble nontemplate-strand ssDNA (positions "-12", "-11", and "-7"; Figs. 3–5; Supplementary Fig. 4), and (4) RNAP recognizes the CRE of a $\sigma^L$-dependent promoter by stacking a nucleotide on a tryptophan and by unstacking, flipping, and inserting a nucleotide into a protein pocket (positions +1 and +2; Figs. 3 and 4; Supplementary Fig. 5). Our biochemical results confirm the functional significance of the observed protein–DNA interactions with the -10 element and CRE of a $\sigma^L$-dependent promoter (Fig. 6a; Supplementary Fig. 6A), provide consensus sequences for the -10 element and CRE of a $\sigma^L$-dependent promoter (Fig. 6b; Supplementary Fig. 6B), and define individual specificity-determining amino-acid–base interactions for two positions of the -10 element of a $\sigma^L$-dependent promoter (positions "-12" and "-11"; Fig. 6c, d). The results provide an indispensable foundation for understanding the structural and mechanistic basis of ECF-σ-factor-dependent transcription initiation.

Our results regarding the connector between σR2 and σR4 of an ECF σ factor, in conjunction with previous results, indicate that all classes of bacterial σ factors contain structural modules that enter the RNAP active-center cleft to interact with template-strand ssDNA and then leave the RNAP active-center cleft by threading through the RNAP RNA-exit channel, providing mechanisms to facilitate de novo initiation, to coordinate extension of the nascent RNA with abortive initiation and initial-transcription pausing, and to coordinate entry of RNA into the RNA-exit channel with promoter escape. For ECF σ factors, as shown here, the relevant structural module is the σR2/4 linker (Figs. 3 and 4; Supplementary Fig. 3); for group-1, group-2, and group-3 σ factors, the module is the functionally analogous—but not structurally homologous—σR3/4 linker[2,3,6,7,12–19,22]; and for group-σ54/σN σ factors, the module is the functionally analogous—but not structurally homologous—region II.3 (RII.3)[20,21].

More broadly, our results, in conjunction with previous results, indicate that cellular transcription initiation complexes in *all* organisms—bacteria, archaea, and eukaryotes—contain structural modules that enter the RNAP active-center cleft to interact with template-strand ssDNA and then leave the RNAP active-center cleft by threading through the RNAP RNA-exit channel. In different classes of bacterial transcription initiation complexes, as described in the preceding paragraph, these roles are performed by the functionally analogous—but not structurally homologous—σR2/4 linker, σR3/4 linker, and RII.3[2,3,6,7,12–22]. In archaeal transcription initiation complexes, these roles are performed by the TFB zinc ribbon and CSB, which are unrelated to the σR2/4 linker, σR3/4 linker, and RII.3[47]. In eukaryotic RNAP-I-, RNAP-II-, and RNAP-III-dependent transcription initiation complexes, these roles are performed by the Rrn7 zinc ribbon and B-reader, the TFIIB zinc ribbon and B-reader, and the Brf1 zinc ribbon,

respectively, each of which is unrelated to the σR2/4 linker, σR3/4 linker, and RII.3[48–56]. It is extraordinary that non-homologous, structurally and phylogenetically unrelated, structural modules are used to perform the same roles in different transcription initiation complexes, and is unknown how or why this occurs.

Our results define the protein–DNA interactions that ECF σ factor σ[L] uses to recognize the -10 element of a σ[L]-dependent promoter. The consensus sequence obtained in this work for the 10-element of a σ[L]-dependent promoter, T"−12"-G"−11"-N"−10"-C/A"−9"-A"−8"-G"−7" (Fig. 6b), confirms and extends the literature-consensus sequence[33–36], and the structural data of this work account for specificity at each specified position of the consensus sequence (Figs. 3–5; Supplementary Fig. 4).

Previous work indicates that RNAP-σ[L] holoenzyme prefers a C-G sequence immediately upstream of the -10-element (C"−14"-G"−13" in our numbering system)[33–36]. Further previous work indicates that this preference may be shared by many RNAP-ECF-σ-factor holoenzymes[25,29,32–35,38,57]; for example, at least 8 of 10 Mtb RNAP-ECF-σ-factor holoenzymes—Mtb RNAP-σ[C], -σ[D], -σ[E], -σ[G], -σ[H], -σ[J], -σ[L], and -σ[M] holoenzymes—exhibit this preference[32–36,38]. In this work, we performed crystallization using nucleic-acid scaffolds that did not contain C"−14"-G"−13", and therefore our crystal structures do not definitively account for the preference for C"−14"-G"−13". However, with the assumption that template-strand nucleotides at positions "-14" and "-13" of a σ[L]-dependent transcription initiation complex are positioned similarly to those in a group-1-σ-factor-dependent transcription initiation complex[15–18], our crystal structures suggest that the C-terminal α-helix of σ[L] σR2 (σ[L] residues 78–82) potentially could make direct, specificity-determining contacts with template-strand nucleotides at these positions. A similar mechanism for recognition of C-G immediately upstream of the -10 element has been proposed for the group-3 σ factor E. coli σ[28][58].

Both our structural results and biochemical results point to the special importance of the nontemplate-strand nucleotide at position "-11" ("master nucleotide"; Figs. 3–6; Supplementary Fig. 4A). Our results regarding recognition of the "-11" "master nucleotide" by an ECF σ factor are consistent with the NMR structure of a complex comprising σR2 from the E. coli ECF σ factor σ[E] and a 5 nt oligodeoxyribonucleotide corresponding to part of the nontemplate strand of the -10 element of a σ[E]-dependent promoter[28,59]. The NMR structure showed unstacking, flipping, and insertion into a protein pocket of the "-11" "master nucleotide" (a cytidine, rather than a guanosine, reflecting the different specificities of E. coli σ[E] vs. Mtb σ[L])[28,59]. The NMR structure did not show unstacking and flipping of the nucleotide at position "-7", reflecting the fact that the oligodeoxyribonucleotide in the NMR structure did not extend to position "-7"[28,59]. The NMR structure also did not show unstacking of the nucleotide at position "-12"[28,59], possibly reflecting an uncertainty in the NMR structure, or possibly reflecting a difference between E. coli σ[E] and Mtb σ[L] in recognition of position "-12".

Based on the NMR structure, Campagne et al.[28,59] hypothesized that the loop of σR2 that forms the protein pocket into which the "-11" "master nucleotide" is inserted—"loop L3" (residues 63-72 of E. coli σ[E], which correspond to residues 56-67 of Mtb σ[L])—serves as a functionally independent, modular determinant of specificity at the "master-nucleotide" position, such that different loop-L3 sequences confer different specificities at the "master-nucleotide" position, in each case, through interactions with an unstacked, flipped, and inserted "master nucleotide". Campagne et al.[28,59] supported this hypothesis by identifying examples of L3-loop sequences that conferred specificity for cytidine, thymidine, and adenosine at the "master-nucleotide" position, and by providing evidence that swapping

L3-loop sequences swaps specificity at the "master-nucleotide" position. Our results provide further support for the hypothesis by identifying an example of an L3-loop sequence, the Mtb σ[L] loop-L3 sequence, that confers specificity for guanosine at the "master-nucleotide" position, and by documenting that specificity for guanosine involves interactions with an unstacked, flipped, and inserted "master nucleotide" (Figs. 3–5; Supplementary Fig. 4A).

In the crystal form identified and analyzed in this work, σR2 of each molecule of transcription initiation complex makes no interactions with other molecules of transcription initiation complex in the crystal lattice (Supplementary Fig. 7A), and, therefore, with this crystal form, it may be possible to substitute σR2 without losing the ability to form crystals (Supplementary Fig. 7A). This potentially provides a platform for systematic structural analysis of σR2 and σR2-DNA interactions for the 13 Mtb σ factors, by determination of crystal structures of transcription initiation complexes containing chimeric σ factors[57,60] comprising σR2 of a Mtb σ factor of interest fused to the σR2/4 linker through σR4 of Mtb σ[L] (Supplementary Fig. 7B; left red arrow) and containing the promoter sequence for the Mtb σ factor of interest. In the crystal form identified and analyzed in this work, there also are no lattice interactions for the connector between σR2 and σR4, and it appears likely there would be no lattice interactions even if that connector were to contain σR3 and a σR3/4 linker, as in group-1, group-2, and group-3 σ factors (Supplementary Figure 7A). Accordingly, this crystal form potentially provides a platform for systematic structural analysis not only of σR2 and its protein-DNA interactions, but also of the connector between σR2 and σR4 and its protein-DNA interactions, for the 13 Mtb σ factors, by determination of crystal structures of transcription initiation complexes containing chimeric σ factors comprising σR2 and the connector of one Mtb σ factor fused to σR4 of Mtb σ[L] (Supplementary Figure 7B, right red arrow) and containing the promoter sequence for the Mtb σ factor of interest.

## Methods

**M. tuberculosis RNAP core enzyme.** Mtb RNAP core enzyme was prepared by co-expression of genes for Mtb RNAP β′ subunit, RNAP β subunit, N-terminally decahistidine-tagged RNAP α subunit, and RNAP ω subunit in E. coli [plasmids pACYC-rpoA, pCOLADuet-rpoB-rpoC, and pCDF-rpoZ[61] (gift of J. Mukhopadhyay, Bose Institute, Kolkata, India); E. coli strain BL21(DE3) (Invitrogen)], followed by cell lysis, polyethylenimine precipitation, ammonium sulfate precipitation, immobilized-metal-ion affinity chromatography on Ni-NTA agarose (Qiagen), and anion-exchange chromatography on Mono Q (GE Healthcare), as described[19].

**M. tuberculosis RNAP σ[A].** Mtb RNAP σ[A] was prepared by expression of a gene for N-terminally hexahistidine-tagged Mtb σ[A] in E. coli [plasmid pET30a-Mtb-σ[A] [62] (gift of S. Rodrigue, Universitè de Sherbrooke, Canada); E. coli strain BL21(DE3) (Invitrogen)], followed by cell lysis, immobilized-metal-ion affinity chromatography on Ni-NTA agarose (Qiagen), and anion-exchange chromatography on Mono Q (GE Healthcare), as described[19].

**M. tuberculosis σ[L].** Mtb RNAP σ[L] was prepared by expression of a gene for N-terminally hexahistidine-tagged Mtb σ[L] in E. coli, followed by cell lysis, immobilized-metal-ion affinity chromatography on Ni-NTA agarose (Qiagen), and anion-exchange chromatography on Mono Q (GE Healthcare). E. coli strain BL21(DE3) (Invitrogen) was transformed with plasmid pSR32[62] (gift of S. Rodrigue, Universitè de Sherbrooke, Canada), encoding N-terminally hexahistidine-tagged Mtb σ[L] under control of the bacteriophage T7 gene 10 promoter. Single colonies of the resulting transformants were used to inoculate 50 ml LB broth containing 50 μg/ml kanamycin, and cultures were incubated 16 h at 37 °C with shaking. Aliquots (10 ml) were used to inoculate 1 L LB broth containing 50 μg/ml kanamycin, cultures were incubated at 37 °C with shaking until OD600 = 0.8, cultures were induced by addition of isopropyl-β-D-thiogalactoside to 1 mM, and cultures were further incubated 16 h at 16 °C. Cells were harvested by centrifugation (4000×g; 15 min at 4 °C), re-suspended in buffer A (10 mM Tris–HCl, pH 7.9, 300 mM NaCl, 5 mM DTT, 0.1 mM phenylmethylsulfonyl fluoride, and 5% glycerol), and lysed using an EmulsiFlex-C5 cell disruptor (Avestin). The lysate was

centrifuged (20,000×g; 30 min at 4 °C), the pellet was re-suspended in buffer B (8 M urea, 10 mM Tris–HCl, pH 7.9, 10 mM MgCl₂, 10 mM ZnCl₂, 1 mM EDTA, 10 mM DTT and 10% glycerol), and the suspension was further centrifuged (20,000×g; 30 min at 4 °C). The supernatant was loaded onto a 5 ml column of Ni²⁺-NTA-agarose (Qiagen) pre-equilibrated in buffer B, and the column was washed with 9 × 15 ml buffer B containing 5, 10, 20, 30, 40, 50, 60, 70, and 80 mM imidazole, and eluted with 50 ml buffer B containing 200 mM imidazole. The sample was subjected to step dialysis for renaturation [10 kDa MWCO Amicon Ultra-15 centrifugal ultrafilters (EMD Millipore); dialysis 4 h at 4 °C against 8 volumes 50% (v/v) buffer C (10 mM Tris–HCl, pH 7.9, 200 mM NaCl, 1 mM DTT, 0.1 mM EDTA, and 5% glycerol) in buffer B; dialysis 4 h at 4 °C against 8 volumes 75% (v/v) buffer C in buffer B; dialysis 4 h at 4 °C against 8 volumes 87.5% (v/v) buffer C in buffer B; and dialysis 4 h at 4 °C against 8 volumes buffer C]; further purified by gel filtration chromatography on a HiLoad 16/60 Superdex 200 prep grade column (GE Healthcare) in 20 mM Tris–HCl, pH 8.0, 100 mM NaCl, 5 mM MgCl₂, and 1 mM 2-mercaptoethanol; concentrated to 10 mg/ml in the same buffer using 10 kDa MWCO Amicon Ultra-15 centrifugal ultrafilters (EMD Millipore); and stored in aliquots at −80 °C. Yields were ~5 mg/L, and purities were ~95%.

Alanine-substituted σ^L derivatives were prepared as described above for the preparation of σ^L, but using plasmid pSR32 derivatives constructed using site-directed mutagenesis (QuikChange Site-Directed Mutagenesis Kit; Agilent).

Selenomethionine-substituted σ^L was prepared as described above for the preparation of σ^L, but using production cultures in 2 L SelenoMethionine Medium Base plus Nutrient Mix[63] (Molecular Dimensions) containing 50 μg/ml kanamycin, incubating at 37 °C with shaking until OD₆₀₀ = 0.8, adding L-selenomethionine (Molecular Dimensions) to 0.3 mM and incubating 15 min at 37 °C with shaking, and adding isopropyl-β-D-thiogalactoside to 0.5 mM IPTG and incubating 16 h at 16 °C with shaking.

**M. tuberculosis RNAP σ^A holoenzyme**. Mtb RNAP σ^A holoenzyme was prepared by co-expression of genes for Mtb RNAP β′ subunit, RNAP β subunit, N-terminally decahistidine-tagged RNAP α subunit, RNAP ω subunit, and N-terminally hexahistidine-tagged σ^A in E. coli [plasmids pACYC-rpoA-sigA, pCO-LADuet-rpoB-rpoC, and pCDF-rpoZ[61] (gift of J. Mukhopadhyay, Bose Institute, Kolkata, India); E. coli strain BL21 (DE3) (Invitrogen, Inc.)], followed by use of cell lysis, polyethylenimine precipitation, ammonium sulfate precipitation, immobilized-metal-ion affinity chromatography on Ni-NTA agarose (Qiagen), and anion-exchange chromatography on Mono Q (GE Healthcare), as described[19].

**M. tuberculosis RNAP σ^L holoenzyme**. Mtb RNAP core enzyme and Mtb σ^L or σ^L derivative were incubated in a 1:4 ratio in 20 mM Tris–HCl, pH 8.0, 100 mM NaCl, 5 mM MgCl₂, and 1 mM 2-mercaptoethanol for 12 h at 4 °C. The reaction mixture was applied to a HiLoad 16/60 Superdex S200 column (GE Healthcare) equilibrated in the same buffer, and the column was eluted with 120 ml of the same buffer. Fractions containing Mtb RNAP σ^L holoenzyme were pooled, concentrated to ~10 mg/ml using 30 kDa MWCO Amicon Ultra-15 centrifugal ultrafilters (EMD Millipore), and stored in aliquots at −80 °C.

**Oligonucleotides**. Oligodeoxyribonucleotides (Integrated DNA Technologies) and the pentaribonucleotide 5′-CpUpCpGpA-3′ (TriLink) were dissolved in nuclease-free water (Ambion) to 3 mM and were stored at −80 °C.

**Nucleic-acid scaffolds**. Nucleic-acid scaffolds RPitc5_sp4, RPitc5_sp5, RPitc5_sp6, and RPo_sp6 (sequences in Supplementary Fig. 2) were prepared as follows: nontemplate-strand oligodeoxyribonucleotide (0.5 mM), template-strand oligodeoxyribonucleotide (0.55 mM), and, where indicated, pentaribonucleotide (1 mM) in 40 μl 20 mM Tris–HCl, pH 8.0, 100 mM NaCl, 5 mM MgCl₂, and 1 mM 2-mercaptoethanol, were heated 5 min at 95 °C, cooled to 25 °C in 2 °C steps with 1 min per step using a thermal cycler (Applied Biosystems), and stored at −80 °C.

**Transcription initiation complexes**. Transcription initiation complexes were assembled by mixing 16 μl 50 μM Mtb RNAP σ^L holoenzyme (in 20 mM Tris–HCl, pH 8.0, 75 mM NaCl, 5 mM MgCl₂, and 5 mM dithiothreitol) and 4 μl 0.4 mM nucleic-acid scaffold (previous section) in 5 mM Tris–HCl, pH 7.7, 0.2 M NaCl, and 10 mM MgCl₂, and incubating 1 h at 25 °C.

**Crystallization, cryo-cooling, and crystal soaking**. Robotic crystallization trials were performed for Mtb RPitc5-σ^L_sp6 using a Gryphon liquid handling system (Art Robbins Instruments), commercial screening solutions (Emerald Biosystems, Hampton Research, and Qiagen), and the sitting-drop vapor-diffusion technique (drop: 0.2 μl transcription initiation complex (previous section) plus 0.2 μl screening solution; reservoir: 60 μl screening solution; 22 °C). 900 conditions were screened. Under several conditions, Mtb RPitc5-σ^L_sp6 crystals appeared within 2 weeks. Conditions were optimized using the hanging-drop vapor-diffusion technique at 22 °C. The optimized conditions for Mtb RPitc5-σ^L_sp6 (drop: 1 μl 40 μM Mtb RPitc5-σ^L_sp6 in 20 mM Tris–HCl, pH 8.0, 75 mM NaCl, 5 mM MgCl₂, and 5 mM dithiothreitol plus 1 μl 100 mM sodium citrate, pH 5.5, 200 mM sodium acetate, and 10% PEG4000; reservoir: 400 μl 100 mM sodium citrate, pH

5.5, 200 mM sodium acetate, and 10% PEG4000; 22 °C) yielded high-quality, rod-like crystals with dimensions of 0.4 mm × 0.1 mm × 0.1 mm in 2 weeks (Supplementary Fig. 2). Crystals were transferred to reservoir solution containing 18% (v/v) (2 R,3 R)-(-)-2,3-butanediol (Sigma-Aldrich) and flash-cooled with liquid nitrogen. Analogous procedures were used for Mtb RPitc5-σ^L_sp4, RPitc5-σ^L_sp5, RPitc-σ^L_sp6, [BrU]RPo-σ^L_sp6, and [SeMet15,76] RPo-σ^L_sp6.

**Diffraction data collection**. Diffraction data were collected from cryo-cooled crystals at Argonne National Laboratory beamline 19ID-D and Stanford Synchrotron Radiation Lightsource SSRL-9-2. Data were processed using HKL2000[64]. The resolution cut-off criteria were: (i) I/σ > = 1.0, (ii) CC₁/₂ (highest resolution shell) >0.5.

**Structure determination and structure refinement**. The structure of Mtb RPitc5-σ^L_sp6 was solved by molecular replacement with MOLREP[65] using the structure of Mtb RPo (PDB 5UHA)[19], omitting σ^A and nucleic acids, as the search model. One molecule of RNAP was present in the asymmetric unit. Early-stage refinement included rigid-body refinement of RNAP core enzyme, followed by rigid-body refinement of each subunit of RNAP core enzyme, followed by rigid-body refinement of 38 domains of RNAP core enzyme (methods as described[6]). Electron density for σ^L and nucleic acids was unambiguous, but was not included in models in early-stage refinement. Cycles of iterative model building with Coot[66] and refinement with Phenix[67] then were performed. Improvement of the coordinate model resulted in improvement of phasing, and electron density maps for σ^L and nucleic acids, which were not included in models at this stage, improved over successive cycles. σ^L and nucleic acids then were built into the model and refined in a stepwise fashion. The final model was generated by XYZ-coordinate refinement with secondary-structure restraints, followed by group B-factor and individual B-factor refinement. The final model, refined to $R_{work}$ and $R_{free}$ of 0.19 and 0.23, respectively, was deposited in the PDB with accession code 6DVC (Table 1).

Analogous procedures were used to solve and preliminarily refine structures of Mtb RPitc5-σ^L_sp4, RPitc5-σ^L_sp5, RPitc5-σ^L_sp6, and [BrU]RPo-σ^L_sp6; models of σ^L and nucleic acids then were built into mF₀-DF𝒸 difference maps, and additional cycles of refinement and model building were performed. The final models were deposited in the PDB with accession codes 6DV9, 6DVB, and 6DVD (Table 1).

Analogous procedures were used to solve and preliminarily refine the structure of [SeMet15,76]RPo-σ^L_sp6; selenium anomalous signals then were used to determine positions of σ^L SeMet15 and σ^L SeMet76, and to confirm the register of σ^L protein residues. The final model was deposited in the PDB with accession code 6DVE (Table 1).

Distance cut-offs for assignment of H-bonds and van der Waals interactions were 3.5 and 4.5 Å, respectively.

**Transcription assays**. For transcription experiments in Fig. 6 and Supplementary Figs. 1E and 6, reaction mixtures contained (10 μl): 75 nM Mtb RNAP σ^L holoenzyme or Mtb RNAP σ^L holoenzyme derivative, 25 nM DNA fragment P-N 25-lac [5′-GCCGCC-3′, followed by positions −100 to −1 of bacteriophage T5 N 25 promoter[68], followed by positions +1 to +9 of E. coli P-lac[69], followed by 5′-A GGATCACAATTTCACACAG-3′; prepared by annealing synthetic oligodeoxyribonucleotides, followed by PCR amplification] or DNA fragment P-sigL-lac or single-base-pair-substituted derivative thereof [5′-GCCGCC-3′, followed by positions −100 to −1 of Mtb P-sigL[36–38] or single-base-pair-substituted derivative thereof, followed by positions +1 to +9 of E. coli P-lac[69], followed by 5′-AGGA TCACAATTTCACACAG-3′; prepared by annealing synthetic oligodeoxyribonucleotides, followed by PCR amplification], 100 μM [α³²P]-UTP (0.03 Bq/fmol), 100 μM ATP, and 100 μM GTP in transcription buffer (40 mM Tris–HCl, pH 8.0, 75 mM NaCl, 5 mM MgCl₂, 2.5 mM DTT, and 12.5% glycerol). Reaction components other than DNA and nucleotides were pre-incubated 5 min at 22 °C; DNA was added and reaction mixtures were incubated 5 min at 37 °C; and nucleotides were added and reaction mixtures were further incubated 5 min at 37 °C. Reactions were terminated by addition of 2 μl loading buffer (80% formamide, 10 mM EDTA, 0.04% bromophenol blue, and 0.04% xylene cyanol). Products were heated 5 min at 95 °C, cooled 5 min on ice, and applied to 16% polyacrylamide (19:1 acrylamide:bisacrylamide, 7 M urea) slab gels (Bio-Rad), electrophoresed in TBE (90 mM Tris–borate, pH 8.0, and 0.2 mM EDTA), and analyzed by storage-phosphor scanning (Typhoon: GE Healthcare). Relative transcriptional activities were calculated from yields of full-length RNA products.

Transcription experiments in Supplementary Fig. 1F were performed in the same manner as transcription experiments in Fig. 6 and Supplementary Figs. 1E and 6, but using reaction mixtures containing (10 μl): 600 nM Mtb RNAP σ^L holoenzyme, 400 nM annealed nontemplate and template strands of nucleic-acid scaffolds RPitc5_sp4, RPitc5_sp5, RPitc_sp6, and RPitc5_sp7 (sequences in Supplementary Fig. 2 for RPitc5_sp4, RPitc5_sp5, RPitc_sp6; 5′-CGTGTCAGTA AGCTGTCACGGATGCAGG-3′ and 5′-CCTGCATCCGTGAGTCGAGGG-3′ for RPitc5_sp7), 1 mM [α³²P]-UTP (0.003 Bq/fmol), 1 mM ATP, and 1 mM CTP in transcription buffer.

Transcription experiments in Supplementary Fig. 1G were performed in the same manner as transcription experiments in Fig. 6 and Supplementary Figs. 1E

and 6, but using reaction mixtures containing (10 µl): 75 nM *Mtb* RNAP σ$^L$ holoenzyme, 25 nM annealed nontemplate and template strands of nucleic-acid scaffolds RPitc5_sp4, RPitc5_sp5, RPitc_sp6, and RPitc5_sp7 (sequences as in preceding paragraph), 500 µM GpA (added together with nucleotides), 100 µM [α$^{32}$P]-UTP (0.03 Bq/fmol), and 100 µM CTP in transcription buffer.

Transcription experiments in Supplementary Fig. 3C (left panel and lanes 1–2 in right panel) were performed in the same manner as transcription experiments in Fig. 6 and Supplementary Figs. 1E and 6, but including 500 µM ApA (TriLink) in reaction mixtures (added together with nucleotides).

**Transcript-release assays.** Transcript-release assays[70] (Supplementary Fig. 3B, right panel, lanes 3–4) were performed by carrying out transcription experiments with transcription complexes immobilized on streptavidin-coated magnetic beads, dividing reaction mixtures into supernatants and pellets by magnetic partitioning, and analyzing transcripts in supernatants (released transcripts) and pellets (unreleased transcripts). Reaction mixtures contained (50 µl): 75 nM *Mtb* RNAP σ$^L$ holoenzyme, 25 nM DNA fragment biotin-P-*sigL-lac* immobilized on streptavidin-coated magnetic beads [prepared by mixing 1.25 pmol biotinylated DNA fragment (biotin incorporated at 5′ end of nontemplate-strand oligodeoxyribonucleotide during synthesis) and 0.05 mg Streptavidin MagneSphere Paramagnetic Particles (Promega; pre-washed with 3 × 150 µl transcription buffer) in 100 µl transcription buffer 30 min at 22 °C, and performing three cycles of removal of supernatant by magnetic partitioning followed by re-suspension in 150 µl transcription buffer at 22 °C], 500 µM ApA, 100 µM [α$^{32}$P]-UTP (0.03 Bq/fmol), 100 µM ATP, and 100 µM GTP in transcription buffer. Reaction components other than bead-immobilized DNA, ApA, and NTPs were pre-incubated 5 min at 22 °C; bead-immobilized DNA was added and reaction mixtures were incubated 5 min at 37 °C; and ApA and NTPs were added and incubated 5 min at 37 °C. Reaction mixtures were separated into supernatants and pellets by magnetic partitioning. Supernatants were mixed with 10 µl loading buffer, heated 5 min at 95 °C, cooled 5 min on ice, and analyzed by urea-PAGE and storage-phosphor imaging as in the preceding section. Pellets were washed with 3 × 200 µl transcription buffer at 22 °C; mixed with 50 µl loading buffer, heated 5 min at 95 °C, cooled 5 min on ice, and analyzed by urea-PAGE and storage-phosphor imaging as in the preceding section.

**Data analysis.** Data for transcription activities are means of at least three technical replicates.

**Reporting summary.** Further information on research design is available in the Nature Research Reporting Summary linked to this article.

## Data availability
Atomic coordinates and structure factors for the crystal structures of *Mtb* RPitc5-σ$^L$_sp4, RPitc5-σ$^L$_sp5, RPitc5-sL_sp6, [BrU]RPo-σ$^L$_sp6, and [SeMet15,76]RPo-σ$^L$_sp6 have been deposited in the PDB with accession codes PDB 6DV9, 6DVB, 6DVC, 6DVD, and 6DVE, respectively. The source data underlying Fig. 6a, c–e and Supplementary Figs. 1C, 1E–G, 3C, and 6A are provided as a Source Data File. Other data are available from the corresponding author upon reasonable request.

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

## Acknowledgements

This work was supported by NIH Grant GM041376 to R.H.E. The authors thank J. Mukhopadhyay and S. Rodrigue for plasmids and APS at Argonne National Laboratory and Stanford Synchrotron Radiation Lightsource for beamline access.

## Author contributions

W.L. and M.S.C. prepared RNAP derivatives. W.L., Y.F. and K.D. performed structure determination. W.L., D.D. and S.M. performed sequence analyses and biochemical experiments. R.H.E. designed the study, analyzed data, and wrote the paper.

## Additional information

**Competing interests:** The authors declare no competing interests.

