## [Peer Review File · Nature Communications]

Reviewers' comments:

Reviewer #1 (Remarks to the Author):

In this study, the authors report crystal structures (3.3-3.8 Å resolution) of *Mycobacterium tuberculosis* transcription initiation complexes containing RNA polymerase, the ECF sigma factor sigmaL, and synthetic nucleic acid scaffolds designed to mimic the downstream fork junction of promoter DNA. The structures show that the ECF sigmaL factor, despite its limited sequence homology to group-1 sigma factors, forms analogous contacts with RNAP and promoter nucleic acids, with modifications that explain differences in promoter sequence recognition within the -10 element and core recognition element. Biochemical analysis of promoter sequence point mutants provided functional validation of the observed promoter contacts and yielded a functional consensus sequence for sigmaL -10 element and CRE sequences.

The data are overall convincing and the figures are clear. I recommend that the manuscript be accepted pending minor revisions as detailed below.

Comments:

1. For these structural studies, the authors test a set of "downstream-fork-junction" nucleic acid scaffolds with different lengths of DNA spanning the region between the -10 element and the downstream DNA (Fig. S2). Could the authors please clarify why the "linker" sequences that were tested all differ from the sequence between the P-sigL -10 promoter element and putative start sites?
2. Given the structural results indicating the stability of a complex containing a 6-nt linker and the biochemical data in Figs. S1E, 6, and S6, can the authors comment on the likely start site for this promoter?
3. The structural comparison of transcription initiation complexes containing either sigmaL or sigmaA (Figs. 4 and S3) revealed that the sigma factor would clash with an initial RNA transcript once it reaches a length of 5 nt or 4 nt, respectively. If the clash between the sigma3-4 linker and the nascent RNA is significantly responsible for the energy barrier leading to the production of short abortive products, wouldn't it be expected that the size (or size distribution) of the abortive products would be different between sigmaL and sigmaA complexes? Could the authors please comment on why this isn't the case (Fig. S3C)?
4. The promoter point mutation and transcription experiments confirm consensus sequences that lead to maximum transcriptional activity in vitro (see pp. 12-13). Are the same motifs found within sigmaL-dependent promoters in sequence alignments or motif analysis (compare to Fig. 6B)? For example, is the extended consensus sequence in the promoter -10 region (-12 to -7) observed? The sigmaL CRE consensus sequence identified in the biochemical experiments is similar to the published group-1 sigma factor CRE consensus sequences (p. 13, last paragraph), with a preference for T at position +1 and G at position +2. This is different than the sequence of the P-sigL promoter shown in Fig. S1D. Is the biochemically-determined motif found within the sigmaL-dependent promoters in vivo?
5. In Figure S3C, is it correct that the 3 nt RNA abortive product migrates slower than a 4 nt RNA product?
6. Regarding the transcription assays described in Figures 6, S1, and S6: it isn't clear which transcription products are used for the quantification of relative transcription activities. Do these relative values include both full-length transcription products and abortive products? A sentence should be added to the methods section to clarify.

Minor points:

1. It would be helpful if the pdb accession code for the structure shown in Figure 1B could be included in the legend.
2. There is a typo on page 17 – Rm7 should be replaced with Rrn7.

Reviewer #2 (Remarks to the Author):

This manuscript by Lin *et al.* and Ebricht describes the structure of the first Mtb RNAP-ECF sigma factor-promoter complex. The structural work is complemented by *in vitro* transcription assays and a slew of structure-based mutagenesis studies that test both the key interacting protein residues as well as the importance and identity of the *sigL* promoter nucleotides. The work is very thorough and well done. The data demonstrate that this ECF sigma factor and likely all ECF sigma factors carry out essentially the identical functions of the sigma A protein, including promoter recognition via altering conformation of the -10 promoter region to effect similar protein-DNA interactions, despite having large deletions in the former proteins relative to the latter protein family. Further, the ECF sigma factors do these functions without significant sequence homology as compared to the sigma A proteins. Hence, the authors claim very reasonably that this is an example of functional analogy by non-homologous sequences. Two additional results that arise from this work include the identifications of the nucleotide specificity/preference in the -10 region and the Core Recognition Element. It was interesting to see how this smaller sigma factor is able to read the -11 and -7 bases as well as how the -12 is now important. Thus, overall this study is quite interesting and a large void in our understanding of how ECF sigma factors “work”. Of course, there are some issues that the authors should address in order to help clarify a few issues.

Table 1: Why are the R_{merge} values so high? Is it simply weak data or is there also an anisotropic component to the data? The authors use $CC_{1/2}$, at least in part, to help with data resolution selection. Not everyone agrees that $CC_{1/2}$ is indeed a magical parameter to use when determining where to cut their high resolution data. Did they examine the electron density maps of their 6DVE when using an $I/\sigma(I)$ value greater than one, i.e. greater than what is normally considered “Noise” and of course resulting in a lower nominal resolution? They should examine their 6DVE structure at a lower resolution, e.g., 4 Å, and decide if the electron density is improved or the same and use that information for their analysis.

Figure S1C: The omega subunit is barely visible. Obviously this is a small protein and one would not expect to that subunit to stain as well as the others. However, the authors should comment on the occupancy/stoichiometry of their complexes. Does each complex have a full omega occupancy?

Figure S1E: It is clear that Mtb RNAP sigmaA has a huge preference for the P-*N25*-*lac* promoter over the P-*sigL*-*lac* promoter. However, the Mtb RNAP sigmaL appears to only have about a 3.3-fold preference for the P-*sigL*-*lac* promoter over the P-*N25*-*lac* promoter. Is this a common feature of the Mtb ECF sigmas, i.e., the ability to still utilise sigmaA promoters with reasonable ability? Please comment.

Figure 5C. The authors present “loss of contact” experiments in which specific, structure-guided mutations of sigma L are assayed for their relative transcriptional activity. The data are fine and informative. However, the authors extend their results in panels 5D and 5E in which they show the data for the sigmaL-H54A change and the sigmaL-D60A substitution. The latter choice makes sense on the basis of the data in Figure 5C. However, why did not the authors test sigmaL-R50A at position 12? Perhaps, this reviewer just missed something but it would appear that R50 has a greater effect on the relative transcriptional activity at -12. Further, this might require rewording

of the sentence on page 14, paragraph 1, lines 14-16 as both H54 and R50 seem to be important for position -12. Please clarify.

Page 18: The authors discuss making chimera of the different sigma factors of Mtb and using their current crystallisation approaches and structure determinations in order to understand all the sigma factors in this mycobacterium. This would appear to be a very reasonable approach to try. However, even if there is space in their crystal form to allow all the chimeras to fit sterically, these experiments might well fail due to different physical-chemical properties of the sequences. As the authors are very likely aware, sometimes a single change in a protein causes very different packing and solubility effects. The authors are encouraged to do these experiments but should add a cautionary note to the manuscript that this might not work for all the sigma factors.

The authors should state somewhere in the text or in an appropriate figure legend what distance cutoffs they are using when considering a hydrogen bond, ionic interaction, stacking interaction and van der Waals contacts.

Figure 1 is not ideal. This is obviously a difficult figure to show and highlight the key features of such a big complex. First, in Figure 1A (top) sigma R1.1 and sigma R1.2 are not pink and red as described in the figure legend. That should be corrected. Second, the catalytic magnesium is not very clear in Figure 1B (top), especially if the reader is looking at a print version. Further, the nontemplate strand is difficult to trace especially in the left side of both structures. The right sides of each panel are much easier to see. The authors are encouraged to work on making the left views more useable.

The authors do not describe why they solved the structure of the Bromodeoxyuridine containing complex. This should be added.

There are a couple of minor typos in the methods section that the authors should correct:

Page 17, paragraph 2, lines 3-4: the word functionally is used twice and likely should be used only once.

Page 40, paragraph 1, line 11: pH 7.9,, should be pH 7.9,

Reviewer #3 (Remarks to the Author):

Lin et al provide structural insight into *M. tuberculosis* ECF sigma containing transcription complexes using downstream fork junction constructs designed from the σ_L autoregulated promoter sequence. They show that despite lack of sequence conservation between the σ_A 3/4 linker sequence and the σ_L 2/4 linker sequence and a difference in lengths of the elements, regions 2 and 4 from both σ factors occupy the same position on the core scaffold and that both linkers leave the active site cleft and enter the RNA exit channel. From these observations they suggest that the σ_L linker probably also pre-organizes the ss template strand for transcription initiation and that its dislocation could also be linked to promoter escape, as has been proposed for the σ_A transcription complex. The authors also show $\sigma_{R2/-10}$ interactions that are in concordance with previous conclusions drawn from *E. coli* $\sigma_{E/-10}$ interactions further demonstrating that non-template unstacking and base flipping is likely a common mechanism used by all σ factors. They demonstrate common holoenzyme (β subunit) interactions with the core recognition element in their σ_L containing complex. This manuscript makes a nice contribution to understanding how ECF σ 's participate in the transcription initiation process and how the linker regions between key modules play analogous roles in all classes of σ factors. As the authors suggest, this paper has

broader implications for appreciating the functionally analogous roles played by non-homologous elements across all domains of life. I cannot find any aspects of this manuscript particularly deserving of negative criticism but I make a few general comments and ask a couple questions below.

1. One of the questions concerning differences between Group I and Group IV σ 's is the length of the linkers between $\sigma R2$ and $\sigma R4$ and the consequences this might have for the locations of the domains in holoenzyme and for their correspondence to the length between the -10 and -35 promoter elements. In the manuscript it claims the 3/4 σA linker is about 80 Aa long while the σL is about 20 Aa. In reality it seems that the extended portions of the σA and σL linkers are rather similar in magnitude and the figure of ~ 80 residues for the 3/4 linker includes the folded $\sigma R3$. In this respect, there is not too much of a mystery of how both σ 's manage to appropriately interact with similarly spaced promoter elements and how both modules occupy the same location on the core RNAP. The slightly longer extended linker in Group I σ seems to result in marginally longer pathways along the active site cleft and through the exit channel. Are these more or less correct interpretations? If so, perhaps a more appropriate length for the 3/4 linker would be about 28 residues (vs about 20 for the ECF 2/4 linker). It seems from the location of folded $\sigma R3$, it does not contribute significantly to a spatial disparity between the R2 and R4 modules in σA vs σL .

2. Owing to their pathway through the exit channel and their interactions with ss template DNA, it is suggested that the linkers of both Group I and IV σ 's may be playing a role of molecular mimicry (of RNA) or as a molecular place-holder. Out of curiosity, and duly noting the presence of few negatively charged residues in the linker regions, I wonder if the authors feel that the notion of molecular mimicry specifically remains a very robust speculation in their minds.

3. Figure S3C: are the 3 nt and 4 nt abortive product labels inverted?

4. The manuscript is well-written and clear, but a few sentences in the introduction could be improved with editing (note that the copy I am reading lacks line numbers):

pg 3 (last sentence): "As RNA synthesis....". This sentence is a little difficult to read. Perhaps using parentheses rather than commas around that internal clause would help.

pg 5 (1st sentence): " ECF σ factors comprise only a module...." should be "are comprised only of a module....".

RESPONSES TO REVIEWERS

Reviewer #1:

In this study, the authors report crystal structures (3.3-3.8 Å resolution) of *Mycobacterium tuberculosis* transcription initiation complexes containing RNA polymerase, the ECF sigma factor sigmaL, and synthetic nucleic acid scaffolds designed to mimic the downstream fork junction of promoter DNA. The structures show that the ECF sigmaL factor, despite its limited sequence homology to group-1 sigma factors, forms analogous contacts with RNAP and promoter nucleic acids, with modifications that explain differences in promoter sequence recognition within the -10 element and core recognition element. Biochemical analysis of promoter sequence point mutants provided functional validation of the observed promoter contacts and yielded a functional consensus sequence for sigmaL -10 element and CRE sequences.

The data are overall convincing and the figures are clear. I recommend that the manuscript be accepted pending minor revisions as detailed below.

Comments:

1. For these structural studies, the authors test a set of “downstream-fork-junction” nucleic acid scaffolds with different lengths of DNA spanning the region between the -10 element and the downstream DNA (Fig. S2). Could the authors please clarify why the “linker” sequences that were tested all differ from the sequence between the P-sigL -10 promoter element and putative start sites?

We have added text to the legend of Fig. S2:

"(sequences in blue designed based on -10 element of P-*sigL* promoter; sequences in pink and red designed based on nucleic-acid scaffold used for determination of structure of *Mtb* RPitc- σ^A in Lin et al., 2017)"

2. Given the structural results indicating the stability of a complex containing a 6-nt linker and the biochemical data in Figs. S1E, 6, and S6, can the authors comment on the likely start site for this promoter?

We have added text to the legend of Fig. S1F and S1G:

"(sequences in Fig. S2; transcription start in each case at template-strand position 2 nt upstream of nucleic-acid-scaffold dsDNA segment)"

"(sequences in Fig. S2; GpA-dependent transcription start in each case at template-strand position 3 nt upstream of nucleic-acid-scaffold dsDNA segment)"

3. The structural comparison of transcription initiation complexes containing either sigmaL or sigmaA (Figs. 4 and S3) revealed that the sigma factor would clash with an initial RNA transcript once it reaches a length of 5 nt or 4 nt, respectively. If the clash between the sigma3-4 linker and the nascent RNA is significantly responsible for the energy barrier leading to the production of short abortive products, wouldn't it be expected that the size (or size distribution) of the abortive products would be different

between sigmaL and sigmaA complexes? Could the authors please comment on why this isn't the case (Fig. S3C)?

We have added text to the legend of Fig. S3C and have added two references:

"consistent with previous results suggesting that abortive-product distributions are determined primarily by the initial transcribed sequence (Hsu et al., 2006; Skancke et al., 2015)"

"Hsu, L., Cobb, I., Ozmore, J., Khoo, M., Nahm, G., Xia, L., Bao, Y., and Ahn, C. (2006). Initial transcribed sequence mutations specifically affect promoter escape properties. *Biochemistry*. 45, 8841-8854."

"Skancke, J., Bar, N., Kuiper, M., and Hsu, L. (2015). Sequence-dependent promoter escape efficiency is strongly influenced by bias for the pretranslocated state during initial transcription. *Biochemistry* 54, 4267-4275."

4. The promoter point mutation and transcription experiments confirm consensus sequences that lead to maximum transcriptional activity in vitro (see pp. 12-13). Are the same motifs found within sigmaL-dependent promoters in sequence alignments or motif analysis (compare to Fig. 6B)? For example, is the extended consensus sequence in the promoter -10 region (-12 to -7) observed? The sigmaL CRE consensus sequence identified in the biochemical experiments is similar to the published group-1 sigma factor CRE consensus sequences (p. 13, last paragraph), with a preference for T at position +1 and G at position +2. This is different than the sequence of the P-sigL promoter shown in Fig. S1D. Is the biochemically-determined motif found within the sigmaL-dependent promoters in vivo?

We have added text on page 12 regarding the -10-element consensus sequence, and we have added text on pages 13-14 regarding the CRE consensus sequence:

"Three of four characterized *Mtb* σ^L -dependent promoters match the consensus sequence of Fig. 6B at position "-11" (P-*sigL*, P-*pks10*, and P-Rv1139c; Hahn et al., 2005; Dainese et al., 2006); two of four match at newly defined position "-8" (P-*sigL* and P-*pks10*; Hahn et al., 2005; Dainese et al., 2006), and two of four match at newly defined position "-7" (P-*sigL* and P-Rv1139c; Hahn et al., 2005; Dainese et al., 2006)."

"Three of four characterized *Mtb* σ^L -dependent promoters match the CRE consensus sequence of Fig. S6B at position +2, the position at which a guanosine is unstacked, flipped, and inserted into a protein pocket (P-*sigL*, P-*pks10*, and P-Rv1139c; Hahn et al., 2005; Dainese et al., 2006)."

5. In Figure S3C, is it correct that the 3 nt RNA abortive product migrates slower than a 4 nt RNA product?

The assignment of 3 nt and 4 nt RNA abortive products is correct. We point out the following text in the legend to Fig. S3C:

"[ApApU and ApApUpU; identities confirmed by reference to products of parallel reactions omitting ATP and GTP; identities further confirmed by reference to products of parallel reactions with *E. coli* RNAP σ^{70} (see Borowiec and Gralla, 1985)]"

6. Regarding the transcription assays described in Figures 6, S1, and S6: it isn't clear which transcription products are used for the quantification of relative transcription activities. Do these relative values include both full-length transcription products and abortive products? A sentence should be added to the methods section to clarify.

We have added text on page 47:

"Relative transcriptional activities were calculated from yields of full-length RNA products."

Minor points:

1. It would be helpful if the pdb accession code for the structure shown in Figure 1B could be included in the legend.

We have added PDB accession codes to the legend of Fig. 1B ("PDB 5UH8" and "PDB 6DVC").

2. There is a typo on page 17 – Rm7 should be replaced with Rrn7.

We have corrected the typo.

Reviewer #2:

This manuscript by Lin et al. and Ebricht describes the structure of the first Mtb RNAP-ECF sigma factor-promoter complex. The structural work is complemented by in vitro transcription assays and a slew of structure-based mutagenesis studies that test both the key interacting protein residues as well as the importance and identity of the sigL promoter nucleotides. The work is very thorough and well done. The data demonstrate that this ECF sigma factor and likely all ECF sigma factors carry out essentially the identical functions of the sigma A protein, including promoter recognition via altering conformation of the -10 promoter region to effect similar protein-DNA interactions, despite having large deletions in the former proteins relative to the latter protein family. Further, the ECF sigma factors do these functions without significant sequence homology as compared to the sigma A proteins. Hence, the authors claim very reasonably that this is an example of functional analogy by non-homologous sequences. Two additional results that arise from this work include the identifications of the nucleotide specificity/preference in the -10 region and the Core Recognition Element. It was interesting to see how this smaller sigma factor is able to read the -11 and -7 bases as well as how the -12 is now important. Thus, overall this study is quite interesting and a large void in our understanding of how ECF sigma factors “work”. Of course, there are some issues that the authors should address in order to help clarify a few issues.

Table 1: Why are the Rmerge values so high? Is it simply weak data or is there also an anisotropic component to the data?

We have added a footnote to Table 1:

^b R_{merge} values for 6DVB, 6DVC, 6DVD, and 6DVE reflect an anisotropic component."

The authors use CC1/2, at least in part, to help with data resolution selection. Not everyone agrees that CC1/2 is indeed a magical parameter to use when determining where to cut their high resolution data. Did

they examine the electron density maps of their 6DVE when using an I/sigma(I) value greater than one, i.e. greater than what is normally considered “Noise” and of course resulting in a lower nominal resolution? They should examine their 6DVE structure at a lower resolution, e.g., 4 Å, and decide if the electron density is improved or the same and use that information for their analysis.

For 6DVE, maps at 3.8 Å ($I/\sigma_I = 1.0$; $R_{merge} > 1.000$; Table 1) show more electron density for sidechains of aromatic and heterocyclic residues than maps at 4.0 Å [$I/\sigma_I = 1.9$; $R_{merge} = 0.937$]. The additional electron density was important for tracing of σ^L .

Figure S1C: The omega subunit is barely visible. Obviously this is a small protein and one would not expect to that subunit to stain as well as the others. However, the authors should comment on the occupancy/stoichiometry of their complexes. Does each complex have a full omega occupancy?

We have added text to the legend of Fig. S1C:

"Staining levels of β' , β , α , ω , and σ^L indicate $\beta':\beta:\alpha:\omega:\sigma^L$ stoichiometry is 1:1:2:1:1. [*Mtb* ω stains weakly (Lin et al., 2017).]"

Figure S1E: It is clear that *Mtb* RNAP sigmaA has a huge preference for the P-N25-lac promoter over the P-sigL-lac promoter. However, the *Mtb* RNAP sigmaL appears to only have about a 3.3-fold preference for the P-sigL-lac promoter over the P-N25-lac promoter. Is this a common feature of the *Mtb* ECF sigmas, i.e., the ability to still utilise sigmaA promoters with reasonable ability? Please comment.

We have added text to the legend of Fig. S1E:

"The magnitude of selectivity of *Mtb* RNAP σ^L holoenzyme for the σ^L -dependent promoter P-sigL (3-fold) is in the range observed for the twelve *Mtb* RNAP ECF σ factors (2- to 30-fold; S.M. and R.H.E., unpublished)."

Figure 5C. The authors present “loss of contact” experiments in which specific, structure-guided mutations of sigma L are assayed for their relative transcriptional activity. The data are fine and informative. However, the authors extend their results in panels 5D and 5E in which they show the data for the sigmaL-H54A change and the sigmaL-D60A substitution. The latter choice makes sense on the basis of the data in Figure 5C. However, why did not the authors test sigmaL-R50A at position 12? Perhaps, this reviewer just missed something but it would appear that R50 has a greater effect on the relative transcriptional activity at -12. Further, this might require rewording of the sentence on page 14, paragraph 1, lines 14-16 as both H54 and R50 seem to be important for position -12. Please clarify.

We have revised the text on "loss-of-contact" experiments to avoid over-stating the results. Specifically, we have replaced "determines specificity" by "contributes to specificity" at each of two places.

Page 18: The authors discuss making chimera of the different sigma factors of *Mtb* and using their current crystallisation approaches and structure determinations in order to understand all the sigma factors in this mycobacterium. This would appear to be a very reasonable approach to try. However, even if there is space in their crystal form to allow all the chimeras to fit sterically, these experiments might well fail due to different physical-chemical properties of the sequences. As the authors are very likely aware,

sometimes a single change in a protein causes very different packing and solubility effects. The authors are encouraged to do these experiments but should add a cautionary note to the manuscript that this might not work for all the sigma factors.

We have revised the text on analysis chimeric σ factors paragraph to avoid overstating the likelihood of success. Specifically, we have replaced "should be possible" by "may be possible," and replaced "there likely would be" by "it appears likely there would be."

The authors should state somewhere in the text or in an appropriate figure legend what distance cutoffs they are using when considering a hydrogen bond, ionic interaction, stacking interaction and van der Waals contacts.

We have added text on page 46:

"Distance cut-offs for assignment of H-bonds and van der Waals interactions were 3.5 Å and 4.5 Å, respectively."

Figure 1 is not ideal. This is obviously a difficult figure to show and highlight the key features of such a big complex.

First, in Figure 1A (top) sigma R1.1 and sigma R1.2 are not pink and red as described in the figure legend. That should be corrected.

We have replaced "pink" and "red" by "light orange" and "orange," respectively, in the legend to Fig. 1A.

Second, the catalytic magnesium is not very clear in Figure 1B (top), especially if the reader is looking at a print version. Further, the nontemplate strand is difficult to trace especially in the left side of both structures. The right sides of each panel are much easier to see. The authors are encouraged to work on making the left views more useable.

We have re-made Fig. 1B to clarify σ , DNA, and the catalytic Mg^{2+} .

The authors do not describe why they solved the structure of the Bromodeoxyuridine containing complex. This should be added.

We point out that the first paragraph of the Results section states:

"For the nucleic-acid scaffold containing a 6 nt spacer, the translocational state of the transcription complex was experimentally verified by preparation of a scaffold having a single 5-bromo-dU substitution and collection of bromine anomalous diffraction data (Table 1; Fig. S2D).

To reiterate the rationales for determination of structures of the 5-bromo-dU- and SeMet-substituted complexes, we have added text to the legends of Figs. S2D and S2E:

"(analyzed to verify the translocational state of the transcription complex)"

"(analyzed to verify the fit of σ^L)"

There are a couple of minor typos in the methods section that the authors should correct:

Page 17, paragraph 2, lines 3-4: the word functionally is used twice and likely should be used only once.

We have corrected the typo.

Page 40, paragraph 1, line 11: pH 7.9., should be pH 7.9,

We have corrected the typo.

Reviewer #3:

Lin et al provide structural insight into M. tuberculosis ECF sigma containing transcription complexes using downstream fork junction constructs designed from the σ_L autoregulated promoter sequence. They show that despite lack of sequence conservation between the σ_A 3/4 linker sequence and the σ_L 2/4 linker sequence and a difference in lengths of the elements, regions 2 and 4 from both σ factors occupy the same position on the core scaffold and that both linkers leave the active site cleft and enter the RNA exit channel. From these observations they suggest that the σ_L linker probably also pre-organizes the ss template strand for transcription initiation and that its dislocation could also be linked to promoter escape, as has been proposed for the σ_A transcription complex. The authors also show $\sigma_{R2/-10}$ interactions that are in concordance with previous conclusions drawn from E. coli $\sigma_{E/-10}$ interactions further demonstrating that non-template unstacking and base flipping is likely a common mechanism used by all σ factors. They demonstrate common holoenzyme (β subunit) interactions with the core recognition element in their σ_L containing complex. This manuscript makes a nice contribution to understanding how ECF σ 's participate in the transcription initiation process and how the linker regions between key modules play analogous roles in all classes of σ factors. As the authors suggest, this paper has broader implications for appreciating the functionally analogous roles played by non-homologous elements across all domains of life. I cannot find any aspects of this manuscript particularly deserving of negative criticism but I make a few general comments and ask a couple questions below.

1. One of the questions concerning differences between Group I and Group IV σ 's is the length of the linkers between σ_{R2} and σ_{R4} and the consequences this might have for the locations of the domains in holoenzyme and for their correspondence to the length between the -10 and -35 promoter elements. In the manuscript it claims the 3/4 σ_A linker is about 80 Aa long while the σ_L is about 20 Aa. In reality it seems that the extended portions of the σ_A and σ_L linkers are rather similar in magnitude and the figure of ~80 residues for the 3/4 linker includes the folded σ_{R3} . In this respect, there is not too much of a mystery of how both σ 's manage to appropriately interact with similarly spaced promoter elements and how both modules occupy the same location on the core RNAP. The slightly longer extended linker in Group I σ seems to result in marginally longer pathways along the active site cleft and through the exit channel. Are these more or less correct interpretations? If so, perhaps a more appropriate length for the 3/4 linker would be about 28 residues (vs about 20 for the ECF 2/4 linker). It seems from the location of folded σ_{R3} , it does not contribute significantly to a spatial disparity between the R2 and R4 modules in σ_A vs σ_L .

We have revised the text on page 7 to avoid overstating the difference in linker length. Specifically, we have replaced "much smaller" by "smaller"; we have replaced "20 residues vs. 84 residues" by "20

residues vs. 84 residues if one includes $\sigma R3$ " and "20 residues vs. 28 residues if one does not include $\sigma R3$ "; and we have deleted "remarkably."

2. Owing to their pathway through the exit channel and their interactions with ss template DNA, it is suggested that the linkers of both Group I and IV σ 's may be playing a role of molecular mimicry (of RNA) or as a molecular place-holder. Out of curiosity, and duly noting the presence of few negatively charged residues in the linker regions, I wonder if the authors feel that the notion of molecular mimicry specifically remains a very robust speculation in their minds.

We have revised the text on page 9 to avoid suggesting that the hypothesized molecular mimicry is an established fact. Specifically, we have replaced "serves as a molecular mimic" by "appears to serve as a molecular mimic."

3. Figure S3C: are the 3 nt and 4 nt abortive product labels inverted?

The assignment of 3 nt and 4 nt RNA abortive products is correct. We point out the following text in the legend to Fig. S3C:

*"[ApApU and ApApUpU; identities confirmed by reference to products of parallel reactions omitting ATP and GTP; identities further confirmed by reference to products of parallel reactions with *E. coli* RNAP σ^{70} (see Borowiec and Gralla, 1985)]"*

4. The manuscript is well-written and clear, but a few sentences in the introduction could be improved with editing (note that the copy I am reading lacks line numbers):

pg 3 (last sentence): "As RNA synthesis...". This sentence is a little difficult to read. Perhaps using parentheses rather than commas around that internal clause would help.

We have replaced the commas with dashes.

pg 5 (1st sentence): " ECF σ factors comprise only a module..." should be "are comprised only of a module..."

The verb "comprise" is correct. The phrase "comprise...of" never is correct. See guidance on "comprise" and "comprised of" at: <https://grammarist.com/usage/compose-comprise/>; <https://grammar.yourdictionary.com/grammar-rules-and-tips/comprise-vs-compose.html>; <https://en.oxforddictionaries.com/definition/comprise>.

REVIEWERS' COMMENTS:

Reviewer #1 (Remarks to the Author):

Unfortunately, my comment #2 was unclear. The comment was directed toward the start site of the native P-sigL promoter (as shown in schematic in Fig. S1D, where the TSS is shown with a question mark). Clarification of this point would not be required for publication. The text changes made by the authors are fine, except that the phrasing of the Fig. S1F legend should match the more concise text in the response letter rather than the longer version currently in the text.

The authors have addressed the most important open points, and I recommend that the manuscript be accepted for publication.

Reviewer #2 (Remarks to the Author):

The authors have provided very satisfactory answers to the questions raised in the previous review and have added good clarifications and appropriate qualifications where required. Figure 1 is now much clearer. This is a very nice study.